# Glutathione in the nucleus accumbens regulates motivation to exert reward-incentivized effort

Ioannis Zalachoras[1†], Eva Ramos-Fernández[1†], Fiona Hollis[1,2‡], Laura Trovo[3], João Rodrigues[1], Alina Strasser[1], Olivia Zanoletti[1], Pascal Steiner[3], Nicolas Preitner[3], Lijing Xin[4], Simone Astori[1], Carmen Sandi[1*]

[1]Laboratory of Behavioral Genetics (LGC), Brain Mind Institute, École Polytechnique Fédérale de Lausanne (EPFL), Lausanne, Switzerland; [2]Department of Pharmacology, Physiology, and Neuroscience, University of South Carolina School of Medicine, Columbia, United States; [3]Nestlé Institute of Health Sciences, Nestlé Research, Société des Produits Nestlé SA, Vers-chez-les-Blanc, Lausanne, Switzerland; [4]Animal Imaging and Technology Core (AIT), Center for Biomedical Imaging (CIBM), EPFL, Lausanne, Switzerland

*For correspondence:
carmen.sandi@epfl.ch

†These authors contributed equally to this work

Present address: ‡University of South Carolina School of Medicine, Columbia, United States

**Abstract** Emerging evidence is implicating mitochondrial function and metabolism in the nucleus accumbens in motivated performance. However, the brain is vulnerable to excessive oxidative insults resulting from neurometabolic processes, and whether antioxidant levels in the nucleus accumbens contribute to motivated performance is not known. Here, we identify a critical role for glutathione (GSH), the most important endogenous antioxidant in the brain, in motivation. Using proton magnetic resonance spectroscopy at ultra-high field in both male humans and rodent populations, we establish that higher accumbal GSH levels are highly predictive of better, and particularly, steady performance over time in effort-related tasks. Causality was established in *in vivo* experiments in rats that, first, showed that downregulating GSH levels through micro-injections of the GSH synthesis inhibitor buthionine sulfoximine in the nucleus accumbens impaired effort-based reward-incentivized performance. In addition, systemic treatment with the GSH precursor N-acetyl-cysteine increased accumbal GSH levels in rats and led to improved performance, potentially mediated by a cell-type-specific shift in glutamatergic inputs to accumbal medium spiny neurons. Our data indicate a close association between accumbal GSH levels and an individual's capacity to exert reward-incentivized effort over time. They also suggest that improvement of accumbal antioxidant function may be a feasible approach to boost motivation.

## Editor's evaluation

This study demonstrates that the level of glutathione in the nucleus accumbens correlates with effortful behaviors both in humans and in rats. By manipulating enzymes involved in the synthesis of glutathione, the authors provide convincing evidence for the causal involvement of glutathione in this process. The results are surprising and provide an important new dimension by which effortful behaviors are regulated.

## Introduction

Motivation facilitates overcoming the cost of effortful actions to attain desired outcomes (*Chong et al., 2016*) and is key to achievement and well-being (*Lepine et al., 2005*; *Duckworth et al., 2015*;

*Kanfer et al., 2017*). Importantly, there are substantial differences in motivated behavior among healthy individuals, manifested as variations in the engagement in effortful activities and in differences in brain activity (*Knutson et al., 2005*; *Berchio et al., 2019*). Motivational deficits - such as apathy, anhedonia, or anergia- are prevalent in many brain pathologies (*Epstein et al., 2006*; *Zald and Treadway, 2017*; *Pessiglione et al., 2018*; *Salamone et al., 2016*). Therefore, unveiling the neurobiological mechanisms underlying individual differences in motivation can help develop novel interventions to boost effortful performance.

A great deal of work in both rodents and humans highlights the nucleus accumbens (NuAc) - a main component of the ventral striatum - as a critical node of the brain's reward and motivation circuitries (*Pessiglione et al., 2018*; *Salamone et al., 2007*; *Croxson et al., 2009*; *Haber, 2011*; *Schmidt et al., 2012*). Indeed, incentives energize effortful behavior (*Berchio et al., 2019*; *Robinson et al., 2014*) through the recruitment of the NuAc (*Schmidt et al., 2012*; *Knutson et al., 2001*; *Pessiglione et al., 2007*; *Hailwood et al., 2018*; *Soares-Cunha et al., 2016*; *Soares-Cunha et al., 2018*; *Gallo et al., 2018*). Moreover, alterations in NuAc function are implicated in psychopathological conditions that involve motivational deficits, such as depression (*Epstein et al., 2006*; *Hanson et al., 2015*; *Muir et al., 2019*; *Treadway et al., 2014*).

Emerging evidence underscores a key role for accumbal mitochondrial function and metabolism in the regulation of motivated behavior (*Hollis et al., 2015*; *van der Kooij et al., 2018a*; *Strasser et al., 2020*; *Gebara et al., 2021*) and vulnerability to develop stress-induced depressive-like behaviors (*Larrieu et al., 2017*; *Cherix et al., 2020*; *Weger et al., 2020*). This is in line with the established relevance of cellular and tissue energy metabolism to maintain healthy brain function (*Hyder et al., 2002*; *Smith et al., 2002*) and goes beyond to posit a link between variation in the levels of specific accumbal metabolites and motivated behavior. However, knowledge regarding the specific accumbal metabolites involved in effortful performance is still scarce.

Recent work in humans using proton magnetic resonance spectroscopy ($^1$H-MRS) focusing on the components of the glutamate/GABA-glutamine cycle has shown that glutamine levels, and particularly, an increased glutamine-to-glutamate ratio in the NuAc predict both, a higher performance related to task endurance and a lower effort perception in a physical effort-based motivated task (*Strasser et al., 2020*). Glutamine is the main precursor for the synthesis of glutamate (*Walls et al., 2015*), a building block for the production of glutathione (GSH; *Forman et al., 2009*; *Sappington et al., 2016*). GSH is a tripeptide (glutamate–cysteine–glycine) essential for several cellular functions and the most prominent antioxidant in the brain (*Figure 1a*; *Rae and Williams, 2017*; *Zalachoras et al., 2020*). It seems, therefore, plausible to hypothesize that GSH levels in the NuAc may be related to the capacity to exert incentivized effort. This hypothesis is supported by convergent evidence indicating that: (i) GSH plays a major role in the reduction of reactive oxygen species (ROS) (*Rae and Williams, 2017*); (ii) ROS generation is coupled to energy production and cellular activity (*Cobley et al., 2018*); (iii) effortful behavior engages the activation of the NuAc (*Schmidt et al., 2012*; *Pessiglione et al., 2007*; *Hailwood et al., 2018*; *Soares-Cunha et al., 2016*; *Soares-Cunha et al., 2018*; *Gallo et al., 2018*); and (iv) altered GSH levels have been reported in the brain of patients in disorders that course along with motivational impairments (*Lapidus et al., 2014*; *Wang et al., 2019*).

To assess this hypothesis, we first explored the relationship between effort-based motivated performance in humans and accumbal levels of several metabolites, including GSH, were quantified using $^1$H-MRS at 7 Tesla (7T) (*Strasser et al., 2020*). These experiments identified a relationship between accumbal GSH levels and motivation to exert effort. Then, to go beyond correlational evidence, we investigated the relationship between accumbal GSH levels in rats and reward-incentivized performance. To this end, we used a progressive ratio (PR) schedule of reinforcement in an operant task that requires the maintenance of nose-poking under increasing work demands and modulated GSH levels through specific interventions in the NuAc. Finally, aiming at a translational application, we evaluated the efficiency of nutritional supplementation with N-acetyl-cysteine (NAC) - a cysteine precursor that has been documented to increase GSH levels in the brain (*Zalachoras et al., 2020*) - to improve rats' motivated performance and explored potential effects in the excitability of the accumbal circuitry.

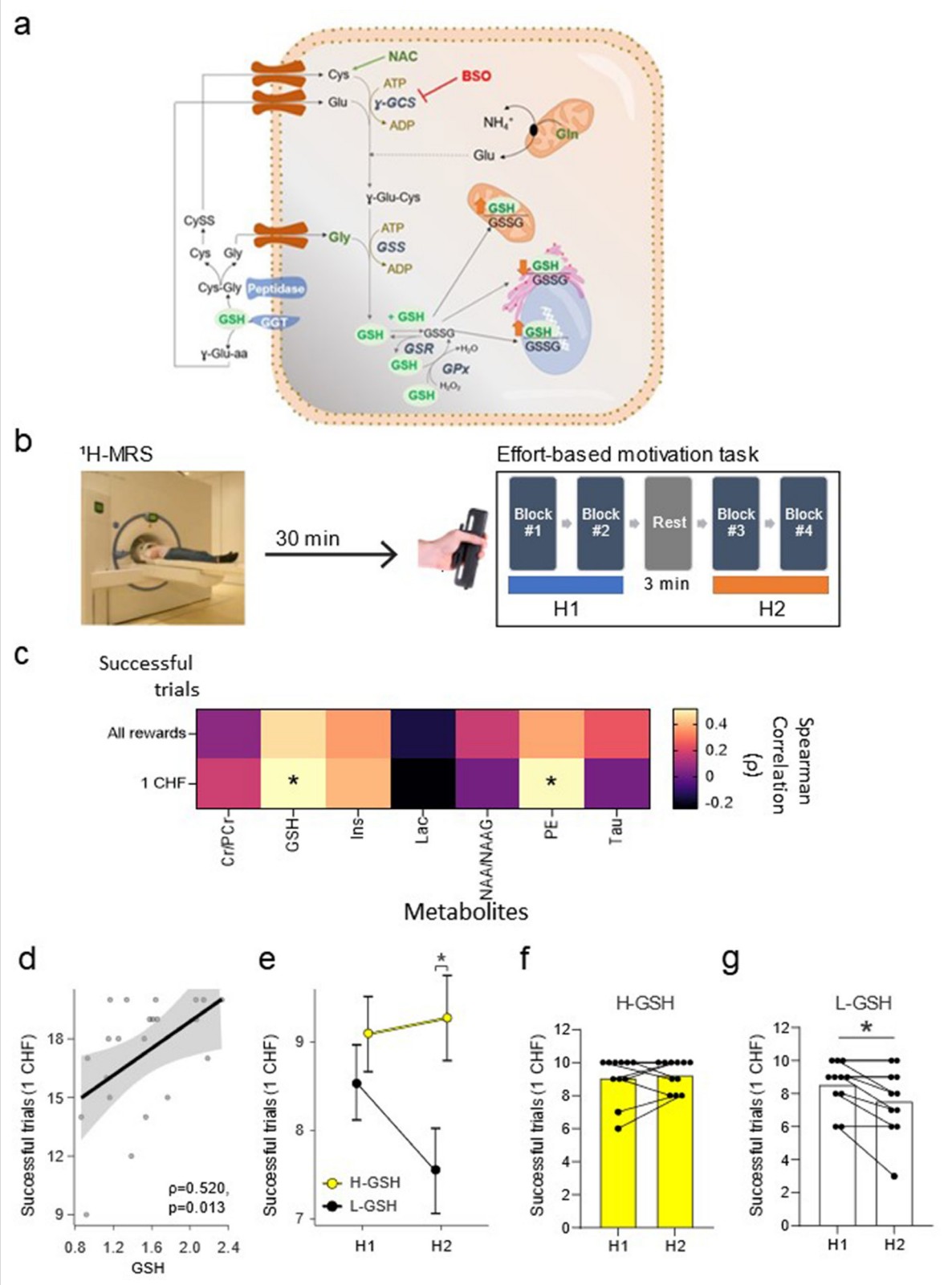

**Figure 1.** Glutathione (GSH) levels in the nucleus accumbens are associated with the number of successful trials on an effort-based motivation task. (**a**) Schematic depiction of GSH metabolic pathways. GSH or γ-L-glutamyl-L-cysteinylglycine, is a tripeptide, that can exist in a reduced (GSH) or in an oxidized (Glutathione disulfide, GSSG) state. It is synthesized in the cytosol by an ATP-dependent two-step process catalyzed by γ-glutamyl cysteine synthase/ligase (γ-GCS) and GSH synthetase (GSS), with γ-GCS as the rate-limiting enzyme. GSH is freely distributed in the cytosol (up to 10 mM

*Figure 1 continued on next page*

*Figure 1 continued*

concentration) and compartmentalized in the nucleus, endoplasmic reticulum, and mitochondria (up to 15%). GSH participates in the neutralization of $H_2O_2$ in the cytosol in collaboration with the glutathione peroxidase (GPx) which takes hydrogens from two GSH molecules, converting one molecule of $H_2O_2$ into two of $H_2O$ and one of GSSG. Glutathione reductase (GSR) reduces GSSG back to GSH using NADPH as an electron donor. The main enzyme that initiates the process of GSH degradation is named glutamyl transpeptidase (GGT). This enzyme is localized in the plasma membrane, facing outward and acting on the extracellular pool of GSH released by the cells. The degradation of GSH by GGTs generates γ-glutamyl-aminoacids and cysteine-glycine dipeptides, which are subjected to the action of membrane-bound peptidase to yield cysteine (Cys) and glycine (Gly). Therefore, the major function of GGT is to recover GSH precursors from extracellular pools which can be taken up by cells and used for the new synthesis of GSH. *N*-acetyl-cysteine (NAC) is a donor of Cys (the rate-limiting substrate), and buthionine sulfoximine (BSO) is a specific inhibitor of γ-GCS. In this work, the systemic treatment of NAC in the drinking water was used to increase GSH levels in nucleus accumbens (NuAc) and the BSO intra-nucleus accumbens infusion to reduce GSH in NuAc. (**b**) Experimental paradigm depiction. The $^1$H-MRS nucleus accumbens scan precedes the effort-based motivation task, which is composed by two halves (H1 and H2), separated by a 3-min rest block. (**c**) Matrix showing all pairwise correlations between metabolites of interest and the number of successful trials in the entire task for all rewards and for 1 CHF rewards. (**d**) Detailed scatterplot for the correlation between GSH levels in the nucleus accumbens and the number of successful 1 CHF trials ( $\rho$ =0.52, p=0.01, corrected p=0.049). (**e**) Participants with lower GSH in the nucleus accumbens (L-GSH) show a lower number of successful 1 CHF trials in the second half of the experiment (**H2**) than those with higher nucleus accumbens GSH levels (H-GSH) (H-GSH=9.27 ± 0.91) successful 1 CHF trials wins, L-GSH=7.55 ± 2.07 successful 1 CHF trials wins; t(20) = −2.45, p=0.02, corrected p=0.04, Cohen's d=1.08. This difference is not present in the first half (H1) (H-GSH=9.09 ± 1.36 successful 1 CHF trials wins, L-GSH=8.55 ± 1.44 successful 1 CHF trials wins; t(20) = −0.91, p=0. 24, corrected p=0.24, Cohen's d=1.08). Participants with high GSH levels in the nucleus accumbens are able to maintain an elevated number of successful 1 CHF trials (**f**) while those with low GSH levels in the nucleus accumbens show a significant reduction of performance in H2 (**g**) (e–f: H-GSH: t(10) = 0.61, p=0.28, corrected p=0.28, Cohen's d=0.19; L-GSH: Wilcoxon W=21.0, p=0.02, corrected p=0.03, rank biserial correlation = 1.00). Significance levels: * p<0.05. Spearman correlation coefficient ( $\rho$ ). N=11/group. See *Figure 1—figure supplement 1* and *Figure 1—source data 1*.

The online version of this article includes the following source data and figure supplement(s) for figure 1:

**Source data 1.** Spectroscopy on human raw data and statistical analysis.

**Figure supplement 1.** Glutathione (GSH) levels in the nucleus accumbens are not associated with the number of successful trials for lower rewards (0.5 CHF and 0.2 CHF).

**Figure supplement 2.** Representative scans and trace from the human occipital lobe.

## Results

### Individual differences in accumbal GSH levels predict effort-based motivated performance in humans

To assess whether the concentration of GSH (for a scheme on GSH metabolic pathways, see *Figure 1a*) in the NuAc in humans is related to motivated performance, and to do so in an unbiased manner, we computed levels of a number of metabolites (creatine and phosphocreatine, GSH, inositol, lactate, N-acetylaspartate and N-acetylaspartylglutamate, phosphatidylethanolamine [PE], and taurine) extracted from the $^1$H-MRS spectra acquired - but not previously analyzed - in the Strasser et al. study (*Figure 1—figure supplement 1a*). Following $^1$H-MRS acquisition at 7T, participants performed an effort-based motivated task in which they could earn different monetary rewards (0.2, 0.5, or 1 CHF, depending on the specific trial) by squeezing a handgrip at 50% of their maximal voluntary contraction and maintaining that force for 3 s (*Figure 1b*). Given that the experiment included two conditions (i.e. performance in isolation or in competition), and performance is influenced by both incentive level and competition (*Strasser et al., 2020*), we hypothesized that levels of relevant (i.e. 'predictive') metabolites would particularly relate to performance evoked by the higher incentive at stake (i.e. 1 CHF) for which incentive value is comparable under both social context conditions (*Strasser et al., 2020*). Indeed, GSH levels in NuAc (but not in the occipital lobe) show a positive correlation with successful 1 CHF - but not total - trials (*Figure 1c*, *Figure 1d*, *Figure 1—figure supplement 1d*, *Figure 1—figure supplement 2*). As hypothesized, there was no correlation between GSH - or any other analyzed metabolites - and the performance at the lower stakes (*Figure 1—figure supplement 1b*). Interestingly, the levels of PE, an abundant glycerophospholipid, also showed a significant correlation with performance for 1 CHF (*Figure 1c* and *Figure 1—figure supplement 1c*); this is an unforeseen finding that will be followed up in future investigations. All these reported associations were significant after correcting for multiple comparisons and controlling for the presence of other metabolites. However, no significant correlations between GSH levels and performance were observed in the occipital cortex (*Figure 1—figure supplement 2* and *Figure 1—source data 1*), considered here as a control brain region.

Then, in order to assess whether GSH levels relate to specific temporal factors in task performance, we split the data from the whole task in two parts (i.e. first and second halves: H1 and H2, each one containing 40 trials; see *Figure 1b*). For the analyses, we divided participants in two groups according to their accumbal GSH content (i.e. low GSH [L-GSH]: N=11; and high GSH [H-GSH]: N=11) based on a median split and grouped rewards into high (1 CHF) and low (0.2 CHF and 0.5 CHF) and tested for the triple interaction of block*reward*GSH level, while controlling for the presence of other metabolites. This factor was significant so we proceeded with two post-hoc analysis to test the block*GSH level interaction for low and high rewards. A significant interaction in the high reward was observed (*Figure 1e and f*), indicating that H-GSH subjects sustained their performance throughout the task, maintaining a similar number of successful 1 CHF trials in the first and second halves of the task. In contrast, L-GSH subjects had a significant reduction in the number of successful trials in the second half of the task as compared to H-GSH subjects (*Figure 1e–g*). These results point to a potential role for accumbal GSH in sustaining performance over time in effortful incentivized tasks.

## Individual differences in accumbal GSH levels in rats are associated with differences in effort-based motivated performance in rats

To determine whether accumbal GSH levels may play a causal role in regulating motivated performance, we switched to rats, which are better suited for mechanistic studies. We performed experiments in males from the outbred Wistar rat strains as our former studies showed a key role for mitochondrial function in the NuAc in motivated behaviors in males from this strain (*Hollis et al., 2015*; *van der Kooij et al., 2018a*; *Gebara et al., 2021*). First, using a parallel approach to our human study, we applied [1]H-MRS at ultra-high field (9.4T) to measure rats' GSH content in the NuAc (*Figure 2a–c*) and divided animals into either H-GSH or L-GSH using a median split, following procedures in our human study (*Figure 2d*). We also measured the levels of GSH in the medial prefrontal cortex (mPFC) by [1]H-MRS in L- and H-GSH rats, observing no statistically significant differences between groups, indicating that the differences observed in the NuAc do not follow the same pattern in all brain regions (*Figure 2—figure supplement 1a-c*). Then, we assessed rats' willingness to exert effort in an aversively motivated task, the forced swim test (FST), in which effort is indicated by the persistence of animal's movement in an inescapable water-filled cylinder (*Proulx et al., 2018*). L-GSH rats spent more time immobile than H-GSH rats when tested in the FST over a period of 15 min (*Figure 2e*), with differences particularly emerging on the second half of the task (*Figure 2f*). The two groups did not differ in anxiety-like behaviors (*Figure 2g*). These results support findings from our human study establishing a link between NuAc GSH levels and motivated performance.

In order to match our mechanistic studies in rats with the human findings reported above, we next tested rats in an effort-based reward-incentivized task, the operant conditioning-based PR. This task evaluates the amount of effort an animal is willing to exert in order to obtain a sucrose pellet as a reward (*Hodos, 1961*). The level of effort at which the animal stops responding to obtain a reward is called the 'breakpoint' (*Stewart, 1975*). Decreases in breakpoint can signal disrupted motivation. Our neuroimaging facility is not equipped with operant boxes, and its sanitary status prevented the rats from being moved after [1]H-MRS testing back to the main animal facility where the behavioral tasks are located. Thus, given this practical constraint, in the next experiments accumbal GSH content was quantified *post-mortem*.

A new cohort of rats was trained in an operant conditioning paradigm, involving a 6-day fixed ratio 1 (FR1) training followed by the PR test (*Figure 3a and b*). GSH was measured using high-performance liquid chromatography (HPLC) following the harvest of the NuAc (*Figure 3C*), which allowed for the establishment of H-GSH and L-GSH groups (*Figure 3d*). The levels of GSH were also measured in the mPFC by HPLC where we did not observe significant differences between groups (*Figure 3—figure supplement 1a*). No significant difference was also found between the groups in anxiety-like behaviors in the elevated plus maze (EPM) and time spent in the center of open field (OF) (*Figure 3—figure supplement 1b, c*), in agreement with our results obtained using [1]H-MRS (*Figure 2g*). The groups also did not differ in locomotion (*Figure 3—figure supplement 1d*; BF01=3.05) or exploration (*Figure 3—figure supplement 1e*; BF01=3.05) as measured in OF and novel object (NO) tests, respectively. Moreover, the two groups did not significantly differ in their performance with a low-effort FR schedule (*Figure 3—figure supplement 1f*), indicating that differences in accumbal GSH levels are not related to differences in learning or in the ability to work for rewards in a low-effort task.

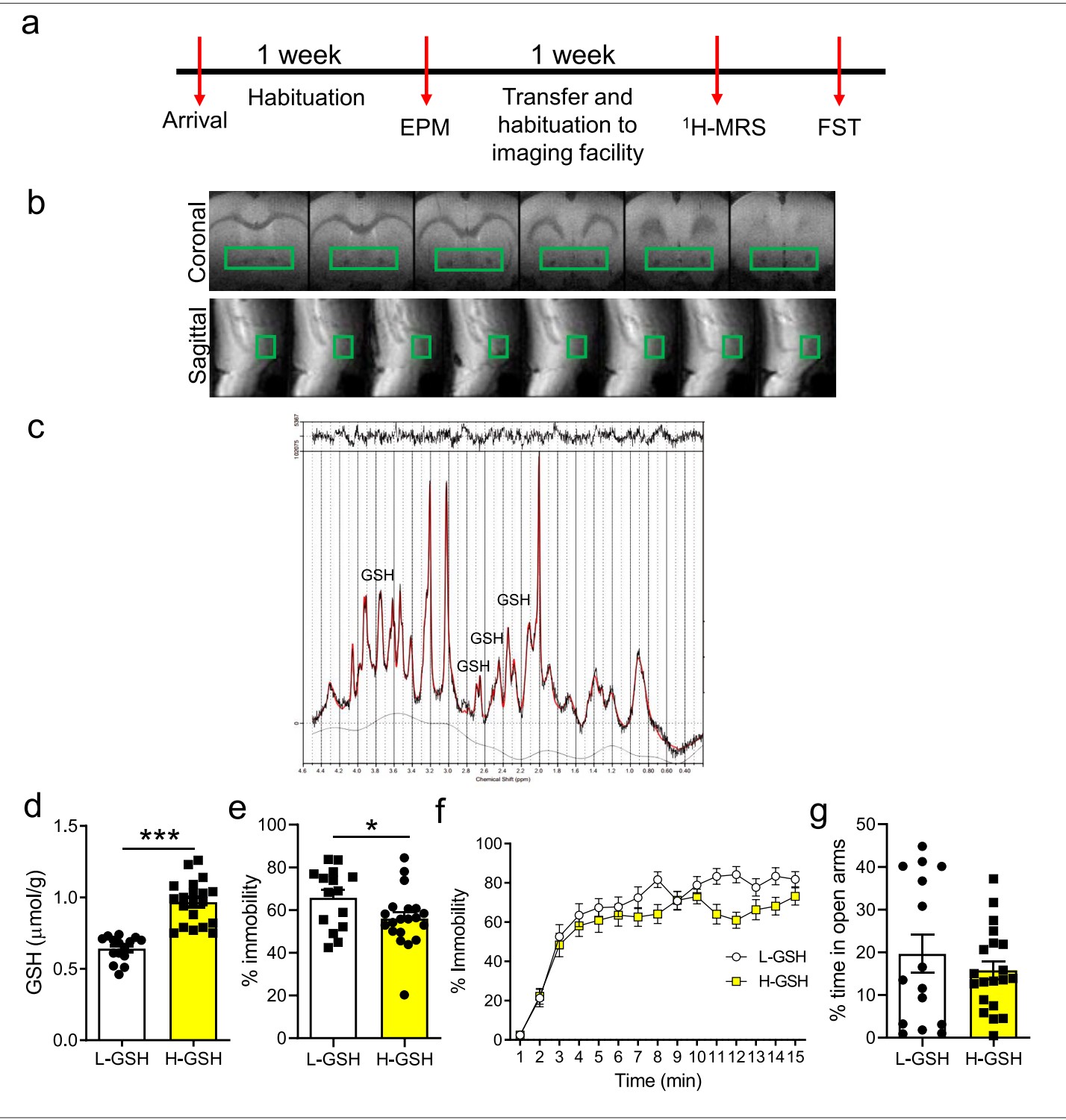

**Figure 2.** Rats differing in accumbal glutathione (GSH) levels show differences in immobility in the forced swim test (FST). (**a**) Experimental design: Upon arrival to the facilities, rats were allowed to acclimatize to the conditions and were handled to habituate to the experimenters. Then they were tested in the elevated plus maze. Subsequently, they were transferred to the imaging facilities and after a further week of habituation, they were scanned in the proton magnetic resonance spectroscopy (¹H-MRS) scanner. A few days after the scanning session, rats were exposed to a 15-min FST and subsequently euthanized. (**b**) Representative scans from the nucleus accumbens, of animals used in the FST. (**c**) Representative trace from the nucleus accumbens. (**d**) A median split was performed to separate rats into two groups, high-GSH (H-GSH) and low-GSH (L-GSH) (two-tailed t-test, t(33) = 7.27, p<0.001). (**e**) H-GSH rats spent less time immobile in the FST (two-tailed t-test, t(33) = 2.10, p=0.04). (**f**) When FST performance was analyzed in 1-min bins, two-way repeated measures ANOVA revealed a significant time × GSH interaction (F(14, 504)=2.10, p=0.01), a marginally non-significant effect of

*Figure 2 continued on next page*

*Figure 2 continued*

GSH (F(1,36)=3.66, p=0.06), and a significant effect of time ([F(36, 504)=11.87, *P*<0.001], N=16–20/group). (**g**) L-GSH and H-GSH did not show significant differences in anxiety (two-tailed t-test, t(33) = 0.85, p=0.40; Bayesian independent samples t-test for the null hypothesis (H0: there is no difference in the % of time in open arms between L-GSH and H-GSH groups) yields a BF01=2.30). See *Figure 2—source data 1*.

The online version of this article includes the following source data and figure supplement(s) for figure 2:

**Source data 1.** Raw data *Figure 2*.

**Figure supplement 1.** Rats differing in accumbal glutathione (GSH) levels show no significant differences in GSH levels in medial prefrontal cortex (mPFC).

However, in the increasing effort-based PR task, H-GSH rats exhibited a higher breakpoint (*Figure 3e*), obtained more rewards (*Figure 3f*), and performed more correct nosepokes (*Figure 3g*) than L-GSH rats. Importantly, the two groups did not significantly differ in the percentage of correct over total (correct and incorrect) nosepokes (*Figure 3—figure supplement 1g*). This indicates that the observed group differences in correct nosepokes are not due to differences in accuracy of responses or in randomly responding to both the correct and incorrect ports, which could have indicated differences in goal-directed behavior. Survival curve analyses showed that a significantly higher number of H-GSH rats reached higher breakpoints (*Figure 3—figure supplement 1h*) and this group also showed a trend toward nosepoking over a longer period of time (*Figure 3—figure supplement 1i*). No group significant differences were observed when analyzing rats' responses over the different ratio requirements (*Figure 3—figure supplement 1j*), but analysis of the number of correct nosepokes over time revealed a significant effect of time and GSH levels (*Figure 3—figure supplement 1k*). In addition to the effort-based reward-incentivized effects observed in the L- and H-GSH groups, NuAc GSH levels were correlated with the number of rewards obtained during the PR task (*Figure 3—figure supplement 1l*). As in humans, our rodent data indicate a role of accumbal GSH in the ability to perform, as well as to persevere, in an effortful incentivized task. Finally, when subsets of H-GSH and L-GSH rats were allowed free access to 20 sucrose pellets (more than any rat managed to acquire during the time they were given access to the PR test), all animals readily consumed all pellets (*Figure 3—figure supplement 1m*), suggesting no group differences in appetite or in the perceived palatability of the pellets. In addition, given the NuAc's involvement in reward processing and anhedonia (*Berridge and Kringelbach, 2008*), we examined whether there were differences in saccharine preference between H- and L-GSH rats under basal conditions but found no significant differences between groups (*Figure 3—figure supplement 1n*).

Taken together, these data indicate that H-GSH rats have a longer active engagement in the PR test than L-GSH rats and their superior performance is not due to a higher rate of responses at the beginning of the PR test. These results indicate a higher endurance capacity to work for rewards over time for rats with higher accumbal GSH levels, resembling the data reported above for human subjects. Therefore, we selected this test and approach to investigate the causal involvement of accumbal GSH in motivated performance.

## Administration of a GSH inhibitor into nucleus accumbens impairs PR test performance

To assess whether decreasing local accumbal GSH levels alters motivated performance, rats were injected with either buthionine sulfoximine (BSO; an inhibitor of gamma-glutamylcysteine synthetase - the rate-limiting enzyme for GSH synthesis, see *Figure 1a* for a schematic of the GSH metabolic pathway) at 7 µg/µl or saline (Sal; the diluent of BSO) intra-NuAc (1 µl per hemisphere; *Figure 4a and b*). Both groups were matched for anxiety, exploratory and locomotor behaviors (before cannulation), and performance during the FR training (*Figure 4—figure supplement 1a-c*). Following training, each group of rats was microinfused with BSO or Sal 24 h prior to the PR test. Our BSO dose was effective in reducing GSH levels as verified via HPLC measurements (*Figure 4c*). In the PR test, BSO-infused animals achieved a lower breakpoint (*Figure 4d*), obtained fewer rewards (*Figure 4e*), and performed fewer correct nosepokes (*Figure 4f*) than the Sal group. No group significant differences were observed in the percentage of correct nosepokes over the total number of nosepokes (*Figure 4—figure supplement 1d*). BSO-treated rats had a lower probability to reach a high breakpoint (*Figure 4—figure supplement 1e*) and a non-significant tendency to perform fewer nosepokes

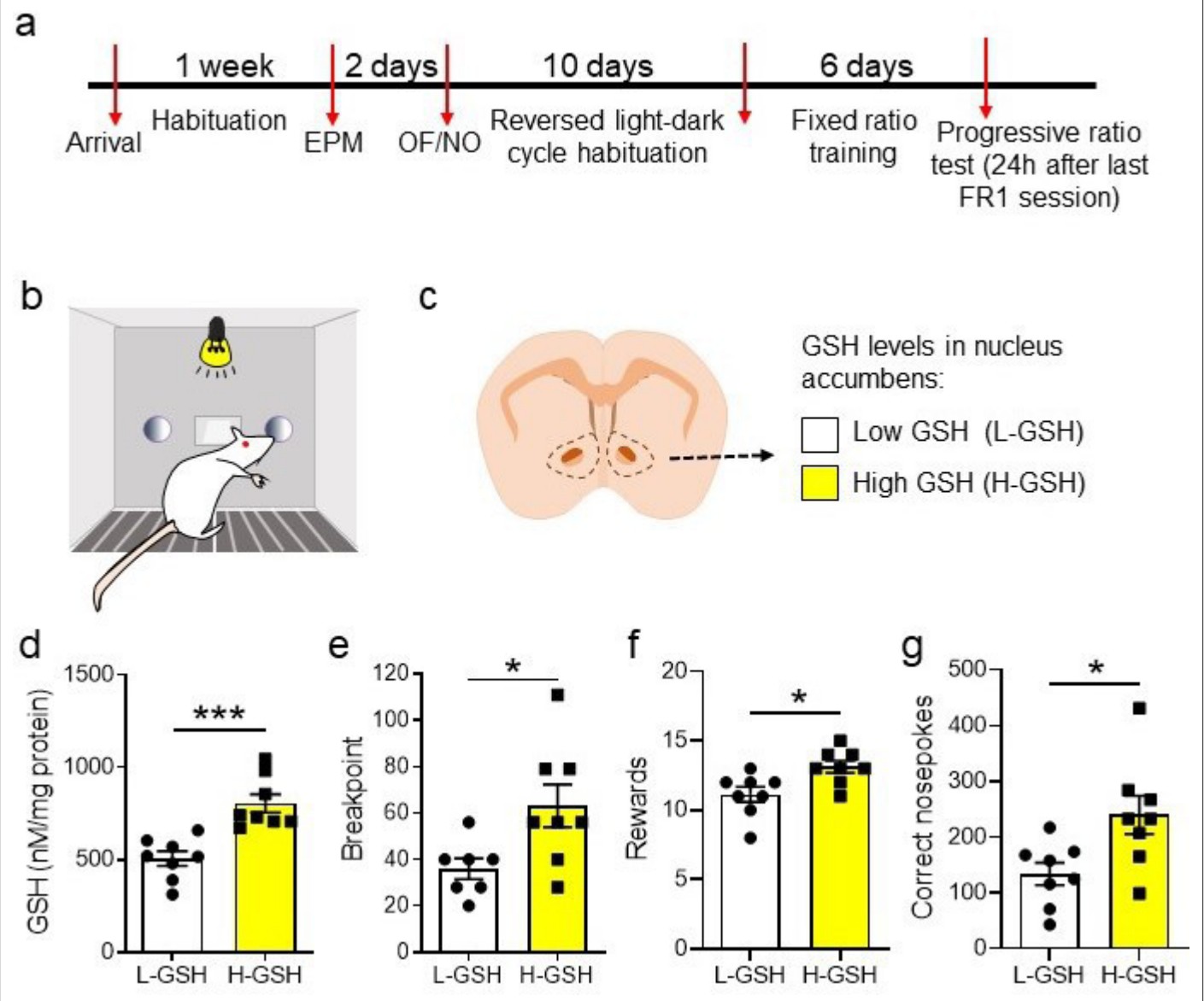

**Figure 3.** High glutathione (H-GSH) levels in nucleus accumbens are associated with an improvement in progressive ratio (PR) test performance. (**a**) Timeline showing the experimental design. After initial acclimatization, handling, and behavioral characterization for anxiety and exploratory behavior, rats were placed in reversed light-dark cycle to habituate for 10 days. Subsequently, rats were trained in the fixed ratio 1 (FR1) paradigm for a total of six sessions (depicted in **b**). Once the PR test was performed, rats were euthanized, their nucleus accumbens was microdissected and snap frozen on liquid nitrogen. Accumbal GSH levels were measured using high-performance liquid chromatography (scheme depicted in **c**). (**d**) Rats were split into two groups based on their accumbal GSH levels (H-GSH and low GSH [L-GSH]), using a median split (t(14) = 4.69, p<0.001) In the PR test, H-GSH rats reached a higher breakpoint (t(14) = 4.69, p<0.001) (**e**), obtained more rewards (t(14) = 2.84, p=0.01) (**f**), and performed more correct nosepokes (t(14) = 2.66, p=0.02) (**g**), indicating that animals with high levels of GSH in nucleus accumbens performed better in the PR paradigm. Unpaired t test two-tailed (***p<0.0001, * p<0.05, N=7–9/group). See *Figure 3—figure supplement 1* and *Figure 3—source data 1*. EPM: elevated plus maze; OF: open field; NO: novel object.

The online version of this article includes the following source data and figure supplement(s) for figure 3:

**Source data 1.** Raw data *Figure 3*.

**Figure supplement 1.** High accumbal glutathione (GSH) levels affect the probability to get a higher breakpoint and the persistence of nosepoking over time.

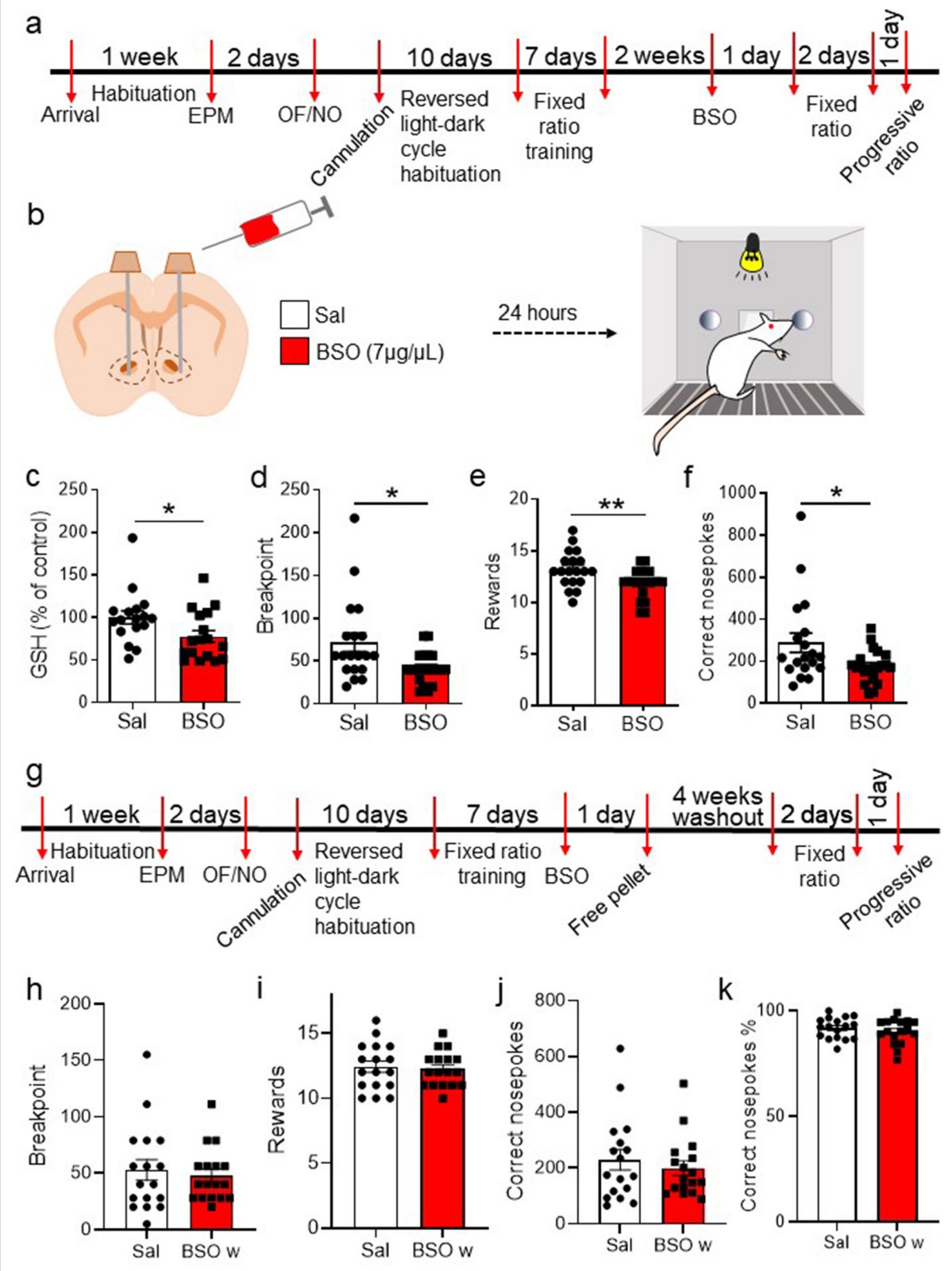

**Figure 4.** Intra-accumbal administration of buthionine sulfoximine (BSO) impairs progressive ratio (PR) performance. (**a**) Cannulated rats were habituated for 10 days in reversed light-dark cycle during their recovery from surgery. Subsequently, they were trained in a fixed ratio 1 schedule for a total of six sessions. (**b**) 24 h before the PR test, BSO or saline (Sal) was infused via a cannula into the nucleus accumbens. (**c**) Glutathione (GSH) levels were measured in the nucleus accumbens of Sal- and BSO-infused rats by high-performance liquid chromatography, demonstrating the ability of BSO

*Figure 4 continued on next page*

*Figure 4 continued*

to reduce GSH levels (unpaired two-tailed t-test t(32) = 2.15, p=0.039 N=17/group). (**d**) PR performance was impaired in BSO-infused animals compared to the vehicle-treated group, as indicated by the lower breakpoint (unpaired two-tailed t-test t(38) = 2.64, p=0.012). (**e**) The number of obtained rewards was also lower following intra-accumbal BSO treatment (unpaired two-tailed t-test t(38) = 2.72, p=0.01). (**f**) BSO-treated rats performed fewer correct nosepokes compared to vehicle-treated rats (unpaired two-tailed t-test t(38) = 2.38, p=0.023, N=19–21/group). (**g**) Timeline showing the experimental design for the washout experiment. Rats treated with vehicle or Sal were tested in the PR task 4 weeks after the BSO infusion. (**h**) The breakpoint was not decreased in BSO-infused rats (unpaired two-tailed t-test, t(32) = 0.47, p=0.64; Bayesian independent samples t-test for the null hypothesis [H0: there is no decrease in the breakpoint in the BSO-infused rats] yields a BF01=4.08, moderate evidence). (**i**) The number of obtained rewards was not significantly decreased in the BSO-infused groups (unpaired two-tailed t-test, t(32) = 0.323, p=0.75; Bayesian independent samples t-test for the null hypothesis [H0: there is no decrease in the obtained rewards in the BSO-infused rats] yields a BF01=3.75, moderate evidence). (**j**) The number of correct nosepokes was not significantly different between groups (unpaired two-tailed t-test, t(32) = 0.68, p=0.50; Bayesian independent samples t-test for the null hypothesis [H0: there is no decrease in the number of correct nosepokes performed by BSO-infused rats] yields a BF01=4.59, moderate evidence). (**k**) Rats previously treated with BSO showed the same percentage of correct nosepokes, over the total number of nosepokes in the washout session (unpaired two-tailed t-test, t(32) = 0.87, p=0.39, N=17 per group; Bayesian independent samples t-test for the null hypothesis [H0: there is no decrease in the percentage of correct nosepokes in the BSO-infused rats] yields a BF01=5.04, moderate evidence). OF: open field; NO: novel object. See *Figure 4—figure supplement 1* and *Figure 4—source data 1*.

The online version of this article includes the following source data and figure supplement(s) for figure 4:

**Source data 1.** Raw data *Figure 4*.

**Figure supplement 1.** Accumbal buthionine sulfoximine (BSO) infusion decreases the probability of reaching higher breakpoints in progressive ratio (PR) task.

(*Figure 4—figure supplement 1f*) than the Sal group. Performance over the different ratio requirements was also not significantly different between the two groups (*Figure 4—figure supplement 1g*). However, BSO-treated rats showed a significantly lower number of correct nosepokes over time (*Figure 4—figure supplement 1h*). Interestingly, a positive correlation between GSH levels and rewards obtained during the task that was observed in control animals was not evident in animals infused with BSO in the NuAc (*Figure 4—figure supplement 1i*).

In order to exclude that the observed BSO effects were due to non-specific actions that could have compromised NuAc integrity or function, we performed several subsequent analyses. At the behavioral level, we tested animals in tasks that engage the NuAc (*Berridge and Kringelbach, 2008*) but do not require effortful behaviors, such as consumption of palatable rewards and operant learning at a low FR1 (*Salamone et al., 2022*). When given free access to 20 sucrose pellets (accounting for the maximum they get access in a PR session) 24 h following either Sal or BSO intra-NuAc infusions (i.e. a time point that corresponds with the timing when animals were tested in the PR following these treatments), animals from both groups did not differ as they all ingested all the pellets within 15 min (*Figure 4—figure supplement 1j*). Then, to assess performance in an FR1 task, we divided animals previously trained and tested in the standard PR task and formerly infused with Sal into two groups, one of them receiving intra-NuAc Sal and the other BSO. In this case, FR1 training was modified as follows: training was performed in different operant boxes, involving levers instead of nosepokes and a tone as discriminative stimulus (instead of house light). We also changed the environment, including a new floor. No significant differences were observed between groups (*Figure 4—figure supplement 1k*), indicating that BSO in the NuAc does not interfere with incentivized operant learning that requires minimum effortful behavior (i.e. FR instead of PR).

Furthermore, in order to rule out a possible toxic effect on the NuAc of the BSO dose used, we performed an activated caspase 3 (a marker of the execution phase of cellular apoptosis) and neuN (a marker of mature neurons) staining assay on accumbal sections from a subset of animals tested in the PR test (*Figure 4—figure supplement 1l-o*). No significant staining differences were observed between Sal and BSO groups, indicating a lack of BSO toxicity at the dose used in this experiment.

To further confirm that the effects of BSO on motivated performance are not due to BSO-induced toxicity or permanent alteration of circuit function, a group of BSO- and Sal-infused animals underwent a 4-week washout period and were then submitted to the PR test (*Figure 4g*). Importantly, 4 weeks after BSO treatment, no decrease on the breakpoint reached was observed in BSO-treated rats (*Figure 4h*; BF01=4.08), the number of obtained rewards (*Figure 4l*; BF01=3.75), or the number of correct nosepokes (*Figure 4j*; BF01=4.75). BSO-washout animals show no decrease in the percentage of correct nosepokes over all nosepokes (both in the active and the inactive port;

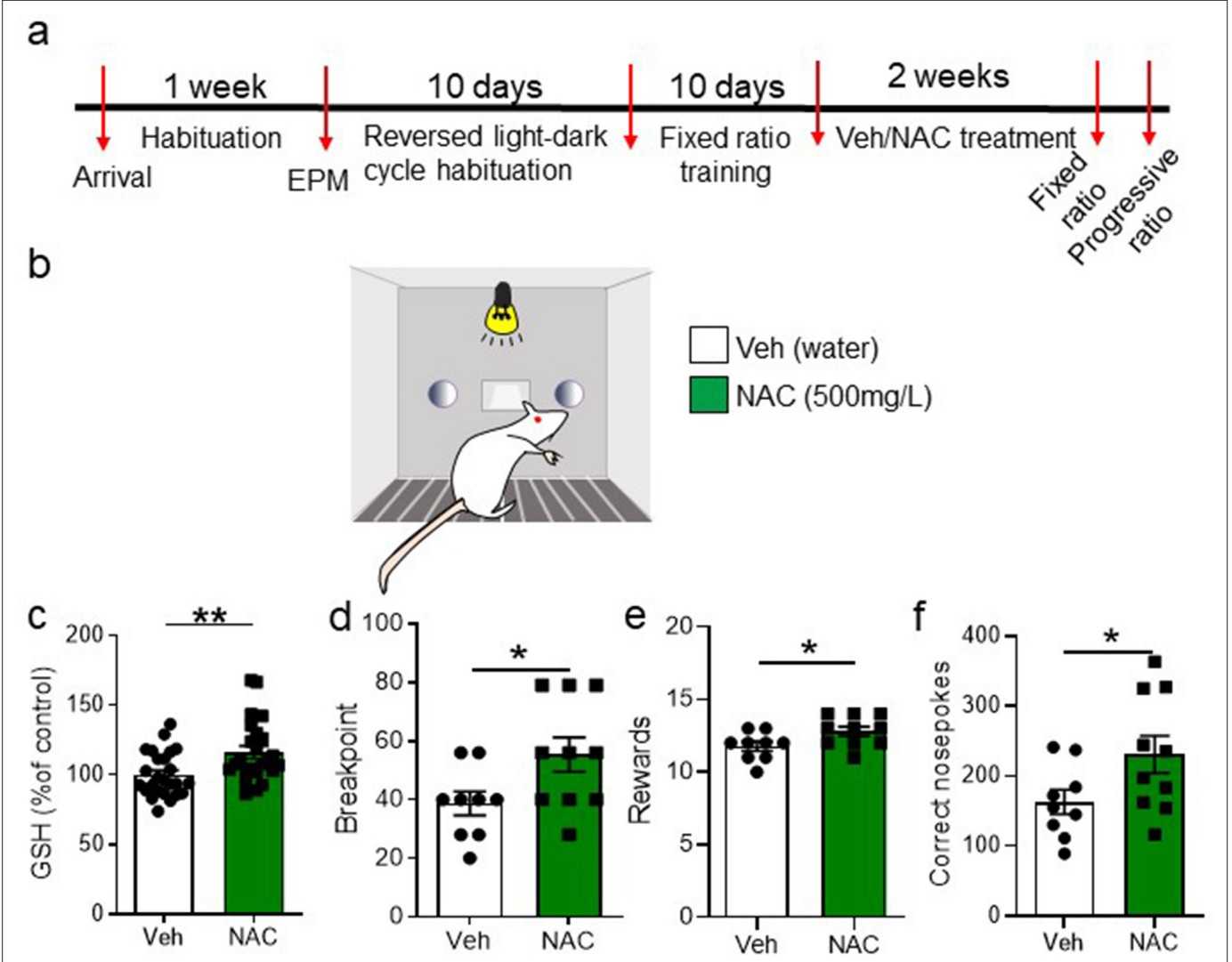

**Figure 5.** N-acetyl-cysteine (NAC) systemic treatment increases glutathione (GSH) levels in nucleus accumbens and ameliorates progressive ratio (PR) task accomplishment. (**a**) Timeline of the experiment. After initial habituation, handling, and behavioral characterization for anxiety, rats were placed in reverse light-dark cycle and trained in an FR1 schedule. Once matched for performance and weight, rats were distributed into vehicle and NAC groups and received normal drinking water or 500 mg/L of NAC in the drinking water, for 2 weeks. Then, both groups were tested in the PR test (**b**). (**c**) A separate cohort of vehicle- or NAC-treated animals was euthanized, and GSH levels were measured in the nucleus accumbens (unpaired two-tailed t-test, t(47) = 2.97, p=0.005, N=24–25/group). (**d**) NAC-treated rats reached a higher breakpoint than vehicle-treated rats (unpaired two-tailed t-test, t(17) = 2.27, p=0.036, N=8/group). (**e**) Similarly, NAC-treated rats obtained more rewards than vehicle-treated rats (unpaired two-tailed t-test, t(17) = 2.21, p=0.041, N=8/group). (**f**) NAC-treated rats performed more correct nosepokes than vehicle-treated rats (unpaired two-tailed t-test, t(15.24)=2.09, p=0.052) (**p<0.01, *p≤0.05, N=9–10/group). See *Figure 5—figure supplement 1* and *Figure 5—source data 1*.

The online version of this article includes the following source data and figure supplement(s) for figure 5:

**Source data 1.** Raw data *Figure 5*.

**Figure supplement 1.** N-acetyl-cysteine (NAC) treatment affects the probability of obtaining a higher breakpoint and persistence of nosepoking over the time but does not change body weight or exploration levels.

*Figure 4k*; BF01=5.04). This set of data further supports the view that lowering accumbal GSH levels leads to a transient impairment in effort-based incentivized performance without inducing toxicity.

## NAC treatment increases accumbal GSH levels and improves PR test performance

Next, we investigated the converse approach, i.e., enhancing GSH synthesis to assess the impact on PR performance. To this end, rats were trained in an FR1 schedule and then divided in two equivalent groups matched according to their performance at training (*Figure 5—figure supplement 1a*), body weight (*Figure 5—figure supplement 1b*), and anxiety levels (*Figure 5—figure supplement 1c-e*). Then, one of the groups received NAC in the drinking water at a concentration of 0.5 mg/L for 2 weeks, while the other group received normal drinking water (vehicle; *Figure 5a and b*). NAC treatment provides a source of cysteine, a rate-limiting building block for GSH synthesis and is known to increase GSH levels in the brain (*Bottino et al., 2021*). Here we confirmed its effectiveness to increase accumbal GSH levels in a separate cohort of animals (*Figure 5c*) in which we also confirmed no drastic changes in basal levels of anxiety-like behaviors (*Figure 5—figure supplement 1c-e*). NAC-treated rats performed better in the PR session than vehicle-treated rats, achieving a higher breakpoint (*Figure 5d*), obtaining more rewards (*Figure 5e*), and performing more correct nosepokes (*Figure 5f*). The percentage of correct nosepokes over the total number of nosepokes (both correct and incorrect) was not different between the two groups (*Figure 5—figure supplement 1g*). Survival analysis showed that a higher percentage of NAC-treated rats reached higher breakpoints (*Figure 5—figure supplement 1h*) and kept nosepoking for longer (*Figure 5—figure supplement 1i*). NAC treatment did not affect rats' performance over the different ratio requirements (*Figure 5—figure supplement 1j*) or the number of correct nosepokes over time (*Figure 5—figure supplement 1k*). Altogether, these data show that a 2-week supplementation with NAC in the drinking bottle increases GSH levels in the NuAc and improves reward-based effortful performance and endurance.

## NAC treatment affects excitatory inputs onto MSNs in a cell-type-specific manner

Next, we asked whether the improvement in behavioral performance induced by NAC treatment was accompanied by changes in the excitability of the accumbal circuitry. The two main NuAc neuronal populations [i.e. medium spiny neurons (MSNs) expressing the dopamine receptor type 1 (D1-MSNs) and type 2 (D2-MSNs)] are classically described as mediators of distinct types of behavior (e.g. approach versus avoidance; *Kravitz et al., 2012*; *Lobo et al., 2010*). In addition, although both core and shell division of the NuAc contribute to motivated behavior, such as performance in the PR task, the core appears to guide goal-directed behavior toward the reward, whereas the shell inhibits goal-irrelevant behavior by avoiding irrelevant, non-rewarded, or less preferable outcome (*Floresco, 2015*). In light of its essential role as initiator of cue-motivated behavior, we focused our investigation on the NuAc core and performed ex vivo patch-clamp recordings from NAC- and vehicle-treated rats combined with biocytin-filling and post-hoc cell-type identification through in situ hybridization (*Figure 6a–c*). No significant differences were observed in passive cell properties, spiking threshold, and firing frequency when cells were depolarized by somatic current injections (*Figure 6—figure supplement 1a-d*), indicating that NAC treatment did not affect intrinsic excitability of MSNs. However, we observed cell-type-specific alterations in the excitatory inputs. Miniature postsynaptic excitatory currents (mEPSCs) recorded in D1-MSNs exhibited significantly larger peak amplitudes in NAC-treated rats, with no significant differences in the inter-event interval (*Figure 6d*). By contrast, mEPSC peak amplitudes were significantly reduced in D2-MSNs, and were also less frequent, as indicated by the pronounced increase in the inter-event interval (*Figure 6e*). Overall, these data indicate that the weight of glutamatergic synaptic drive in the NuAc core is altered by the NAC treatment, such that D1-MSNs receive larger excitatory inputs, while these are overall reduced in D2-MSNs. The resulting augmented engagement of D1-MSNs over D2-MSNs is consistent with the purported roles of the two cell types in approach/avoidance behavior, thus offering a plausible cellular substrate for the enhanced performance in the PR task.

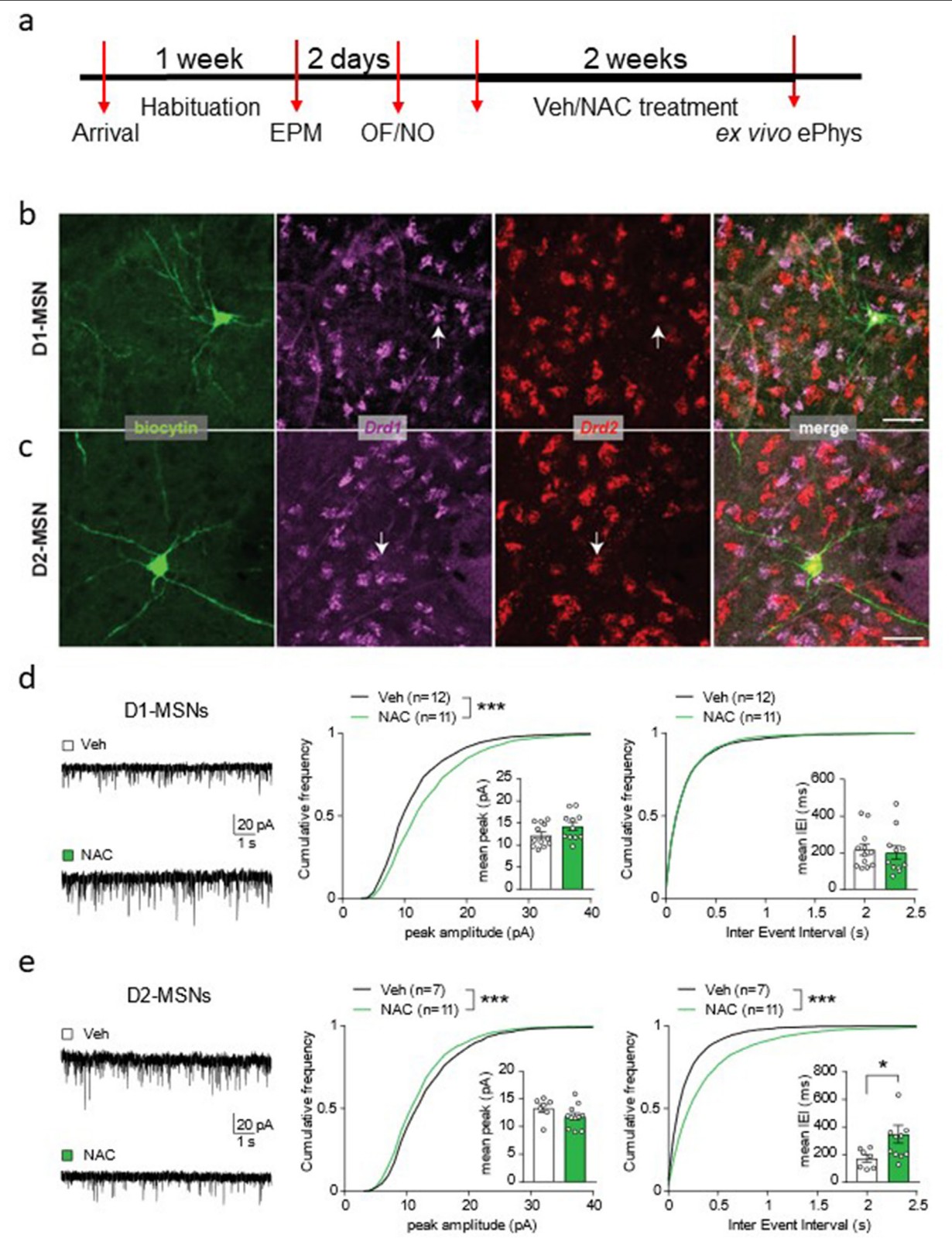

**Figure 6.** N-acetyl-cysteine (NAC) treatment affects excitatory inputs onto medium spiny neurons (MSNs) in a cell-type-specific manner. (**a**) Experimental timeline for ex vivo electrophysiological recordings in NAC-treated rats. (**b–c**) Confocal micrographs showing neurons in the nucleus accumbens core filled with biocytin during patch-clamp recordings and identified post-hoc as dopamine receptor type 1-MSN (D1-MSN) (**b**) and dopamine receptor type 2-MSN (D2-MSN) (**c**) based on RNAscope reactivity for *Drd1* and *Drd2* mRNA (color-coded). Scale bar, 50 μm. (**d**) Left, example traces of miniature

*Figure 6 continued on next page*

*Figure 6 continued*

excitatory postsynaptic currents (mEPSCs) in D1-MSNs from vehicle (Veh)- and NAC-treated rats. Cumulative frequency plots indicate larger peak amplitude in NAC-treated rats (middle panel, Kolmogorov-Smirnov test, D=0.16, ***p<0.001, n=11–12/group), with no change in mEPSC interevent interval (right panel, Kolmogorov-Smirnov test, D=0.018, p=0.84). Insets with bar graphs represent mean values per cell (peak, unpaired t test, t(19.52)=1.69, p=0.11; Bayesian independent samples t-test for the null hypothesis [H0: there is no difference in mean values per cell between groups] yields a BF01=0.96; inter-event interval, Mann-Whitney test, U=56, p=0.57). (e) Left, example traces of mEPSCs in D2-MSNs from Veh- and NAC-treated rats. Cumulative frequency plots indicate smaller peak amplitude (middle panel, Kolmogorov-Smirnov test, D=0.075, ***p<0.001, n=7–11/group) and larger inter-event interval in NAC-treated rats (right panel, Kolmogorov-Smirnov test, D=0.22, ***p<0.001). Insets with bar graphs represent mean values per cell (peak, unpaired t test, t(13.79)=1.59, p=0.14; Bayesian independent samples t-test for the null hypothesis [H0: there is no difference in mean values per cell between groups] yields a BF01=1.08; inter-event interval, Mann-Whitney test, U=16, *p=0.04). See *Figure 6—figure supplement 1* and *Figure 6—source data 1*.

The online version of this article includes the following source data and figure supplement(s) for figure 6:

**Source data 1.** Raw data *Figure 6*.

**Figure supplement 1.** N-acetyl-cysteine (NAC) treatment does not affect medium spiny neuron (MSN) intrinsic excitability.

## Discussion

Here, we establish a novel role for GSH in specific brain structures in motivation. Specifically, we identify GSH levels in the NuAc as critically related to the individual's capacity to keep on exerting effortful behavior over time in reward-incentivized tasks. We first underscored GSH levels in the NuAc as predictive of steady execution in an effort-based monetary incentivized task in humans. Participants with higher accumbal GSH levels were able to steadily maintain good performance until the end of the task, while those with lower levels showed a decay in the second half. To investigate causality, we switched to rats in which we confirmed that individual differences in accumbal GSH levels predict performance in both aversively and appetitively motivated tasks. The involvement of GSH in effort-based motivated performance was supported by the observation of impaired performance in the PR test following downregulation of accumbal GSH levels through the microinjection of the GSH synthesis inhibitor BSO. Furthermore, systemic treatment with NAC, a source of cysteine known to increase brain GSH, led to increases in accumbal GSH levels as well as in rats' breakpoint in the PR task, that was possibly due to a shift in glutamatergic inputs to D1- and D2-MSNs. Our results highlight the ability of accumbal GSH levels to facilitate motivated endurance over time. These findings may open new avenues to consider accumbal GSH as both a biomarker for motivational disturbances and a potential target for therapeutic interventions.

Although we set the study on the specific hypothesis that accumbal GSH may be involved in effort-based motivated performance, our first unbiased analysis considering a range of metabolites measured through [1]H-MRS at 7T in humans yielded GSH as the metabolite specifically predictive of performance out of a total of 10 metabolites analyzed. Given that GSH shows a low physiological concentration and its spectra strongly overlaps with other metabolites, its measurement is challenging, especially at low magnetic fields. Increasing spectral signal-to-noise ratio and resolution at high magnetic field (7T and above) as used here is critical for improving metabolites quantification, especially for J-coupled metabolites such as GSH (*Lin et al., 2012*; *Xin et al., 2013*). Using a similar approach in rats ([1]H-MRS at 9.4T), we confirmed the capacity of accumbal GSH levels to predict performance in an aversively motivated task, and post-mortem analyses related GSH levels to performance in a reward-incentivized task. Specifically, rats with higher accumbal GSH levels spent less time immobile in the FST (i.e. displaying more active coping responses to adversity) and had higher breakpoints in the PR task than their lower GSH counterparts.

Formerly, we identified a predictive value for glutamine, and particularly the glutamine-to-glutamate ratio in motivated performance in humans, and hypothesized the contribution of these metabolites to fuel mitochondrial function under enhanced NuAc engagement during effortful incentivized behaviors. Indeed, human studies have shown that the NuAc (or the ventral striatum more broadly) is the main brain structure to respond to incentives (*Schmidt et al., 2012*; *Knutson et al., 2001*; *O'Doherty et al., 2004*; *Yacubian et al., 2006*) being incrementally recruited as the expected reward increases (*Knutson et al., 2001*; *Sescousse et al., 2015*). NuAc activity relates to effort-based cost-benefit valuation (*Croxson et al., 2009*), acting as the motivating force (*Pessiglione et al., 2007*) that engages both cognitive and motor downstream brain regions (*Schmidt et al., 2012*). These human studies measured blood oxygen level-dependent signals through functional magnetic resonance imaging,

as a proxy of neural activity. Work in rats using *in vivo* oxygen ($O_2$) amperometry similarly showed increased $O_2$ responses to rewards as a function of increased required effort to obtain them. Animals with higher breakpoints showed a greater magnitude of the NuAc $O_2$ responses, suggesting that individual differences in the capacity to remain on task for longer are related to the magnitude of $O_2$ responses in the NuAc (*Hailwood et al., 2018*). Given that the main $O_2$-consuming cells in the brain are neurons (*Hall et al., 2012*), increased $O_2$ in the NuAc mainly represents the energetic cost of neuronal activity. Specifically, oxygen consumption enables to meet increased energy demands in neural circuits by facilitating oxidative phosphorylation in the mitochondria and thus, contributing to the generation of ATP (*Schmidt et al., 2012*; *Hall et al., 2012*; *Schneider et al., 2019*). However, oxygen consumption generates free radicals and other ROS which, when excessive and not sufficiently counteracted by cellular antioxidant systems can produce oxidative stress, toxic to cells and molecules, and to which the brain is particularly susceptible (*Cobley et al., 2018*). Importantly, higher mitochondrial oxygen respiratory capacity in the NuAc in rodents has been shown to positively relate to the duration of active coping/escaping responses in the FST (*Gebara et al., 2021*) and to winning a social competition (*Hollis et al., 2015*; *van der Kooij et al., 2018a*). Accordingly, our findings here suggest that a higher accumbal GSH production may act in concert with mitochondrial oxidative phosphorylation processes to facilitate motivated effort. Its causal contribution to motivated performance was established in our study by the direct inhibition of GSH levels by BSO intra-NuAc injection.

GSH is the major antioxidant as well as a redox and cell-signaling regulator (*Rae and Williams, 2017*; *Dringen, 2000*). Given the high susceptibility of the brain to oxidative stress (*Cobley et al., 2018*), neurons require high levels of GSH to counteract oxidative stress and sustain cellular integrity and circuit function (*Fernandez-Fernandez et al., 2018*). Therefore, our results, in both humans and rats, relating individual differences in accumbal GSH content to the capacity to succeed in task performance over time, support the view that higher individual GSH levels provide an advantage to deal with oxidative processes generated by the drastically enhanced energy demand - involving increased $O_2$ utilization - required in NuAc circuits to maintain performance (see above). However, given our previous findings implicating the glutamine-glutamate ratio in effortful performance (*Strasser et al., 2020*), and recent evidence indicating that GSH may be a physiological reservoir of glutamate neurotransmitter when the glutamate-glutamine shuttle is inhibited (*Sedlak et al., 2019*), the alternative non-mutually exclusive possibility that higher GSH levels may be contributing to the glutamate-glutamine cycle [or vice versa; note that glutamine may also contribute directly to GSH synthesis (*Sappington et al., 2016*)] and, thus, facilitating performance cannot be ruled out.

We also carried out a translational approach and examined the potential of a prolonged treatment with the GSH booster NAC to improve motivation (see *Figure 1a*). Our administration regime effectively increased accumbal GSH content, supporting the efficiency of systemic NAC treatment in targeting antioxidant levels in the ventral striatum. Importantly, NAC treatment was also effective in increasing rats' breakpoint in the PR task, supporting its ability to ameliorate motivational deficiencies. Selective activation of specific glutamatergic inputs to the NuAc has been shown to promote reward-based motivation (*Stuber et al., 2008*; *Day et al., 2007*; *Flagel et al., 2011*). Hence, we provide novel evidence that prolonged systemic treatment with NAC induces cell-subtype-specific changes in synaptic transmission in the NuAc core: quantal glutamatergic currents were larger in D1-MSNs, while the excitatory drive onto D2-MSNs was overall diminished. This suggests that, in NAC-treated rats, excitatory inputs promote activation of D1-MSNs over D2-MSNs as a result of the increased postsynaptic responses in D1-MSNs as well as the concomitant reduction in collateral inhibition provided by D2-MSNs (*Ducrocq et al., 2020*). The shift toward D1-MSN activation is expected to support motivated behavior, according to the canonical roles of D1- and D2-MSNs in positive reinforcement and aversion, respectively (*Kravitz et al., 2012*; *Lobo et al., 2010*; *Floresco, 2015*); but note that several studies have suggested that D2-MSNs may also contribute to reward-based motivation (*Ducrocq et al., 2020*; *Trifilieff et al., 2013*; *Filla et al., 2018*) and that MSN subtype-specific actions may depend on different patterns of activation (*Soares-Cunha et al., 2020*). The mechanisms whereby NAC treatment can affect glutamatergic function in the NuAc are multiple. In addition to promoting GSH synthesis, cystine derived from NAC also binds the cystine-glutamate exchanger in glial cells and shifts the extracellular levels of glutamate (*Tardiolo et al., 2018*). Moreover, alterations in the metabolic pathways leading to the synthesis of GSH can affect glutamate levels and accumulation (*Koga et al., 2011*), and glutamate activity can also be shaped by GSH (*Sedlak et al., 2019*). Future

studies are needed to tackle the molecular mechanisms mediating these cell-type-specific rearrangements. Notably, opposite effects on glutamatergic transmission have been shown to occur in D1- and D2-MSNs upon drug exposure (*Zinsmaier et al., 2022*), confirming that synaptic adaptations in the NuAc can take place in a cell-type-specific manner. NAC interventions have been applied in animal models of addiction and have proven successful in rescuing drug-seeking behavior by normalizing extracellular glutamate levels, thereby modulating accumbal excitatory transmission via metabotropic receptors (*Olive et al., 2012*). However, in contrast to our results, such cellular mechanisms have been addressed in response to acute NAC administration and do not exhibit cell-type specificity. Thus, our findings are likely the result of the potent antioxidant capacity of GSH facilitating the engagement of accumbal circuitry to produce motivated behavior, as well as GSH-independent mechanisms involving more complex and long-term modifications, including a reshaping of synaptic glutamatergic activity.

Variability in accumbal GSH levels can be due to both genetic and non-genetic life and situational factors. Importantly, as shown in some cell types, intra-individual variation in GSH content seems to be rather stable over time (*van 't Erve et al., 2013*). Individual capacity to produce GSH is determined by genetic variability in enzymes involved in its production and/or regeneration, including several common single nucleotide polymorphisms that regulate GSH levels and associated processes (*Dasari et al., 2018*). In addition, several life conditions can also affect brain GSH levels, particularly under situations leading to increased ROS expression and oxidative stress such as during ageing or psychogenic stress (*Zalachoras et al., 2020*; *Sohal et al., 1994*). Accordingly, several psychopathological alterations - such as depression or schizophrenia - in which low GSH levels have been reported are associated with motivational deficits or fatigue (*Lapidus et al., 2014*; *Wang et al., 2019*; *Xin et al., 2016*). Remarkably, we report that inter-individual differences in accumbal GSH levels in healthy, human and rodent populations (i.e. within the normal physiological range) are related to differences in effortful behavior. Therefore, our findings here go well beyond previously reported correlations of brain GSH levels with psychiatric conditions to causally implicate accumbal GSH levels in motivated behavior in healthy individuals. Moreover, we provide a proof of principle that nutritional treatments leading to 15–20% change in accumbal GSH content are able to modulate motivated performance in healthy individuals.

In conclusion, we establish GSH levels in the NuAc both as a predictive marker of differences in reward-based effortful performance and as a potential target for nutritional or other type of interventions. We also provide strong evidence for a promising potential of chronic NAC supplementation to boost accumbal GSH levels and to regulate motivation to exert reward-incentivized effort. However, it is important to note that our sample sizes are relatively small, and thus the absence of significant effects should be interpreted with caution. As our studies were performed in male humans and rodents, it is important to note as a limitation to our findings. We are currently expanding our studies to female populations, where we hope to find similar relationships between GSH and motivated behavior.

## Materials and methods

### Key resources table

| Reagent type (species) or resource | Designation | Source or reference | Identifiers | Additional information |
|---|---|---|---|---|
| Strain, strain background (*Wistar, male*) | Rats | Charles River, L'Arbresle, France | RRID:RGD_2312504 | |
| Antibody | Anti-NeuN (rabbit polyclonal) | Millipore | Cat. #:ABN78 RRID:AB_10807945 | 1:1000 |
| Antibody | Anti-cleaved caspase-3 (rabbit polyclonal) | Cell Signaling Technology | Cat. #:9661 RRID:AB_2341188 | 1:200 |
| Antibody | Anti-mouse Alexa Fluor 488 (goat polyclonal) | Thermo Fisher Scientific | Cat. #:A-11029 RRID:AB_2534088 | 1:1000 |
| Antibody | Anti-rabbit Alexa Fluor 488 (donkey polyclonal) | Thermo Fisher Scientific | Cat. #:A-21206 RRID:AB_2535792 | 1:1000 |

*Continued on next page*

*Continued*

| Reagent type (species) or resource | Designation | Source or reference | Identifiers | Additional information |
|---|---|---|---|---|
| Peptide, recombinant protein | Streptavidin, Alexa Fluor 488 conjugate | Thermo Fisher Scientific | Cat. #:S-11223 | 1:500 |
| Commercial assay or kit | RNAscope Fluorescent Multiplex Reagent Kit | ACD, Bio-Techne | Cat. #:320850 | probes: Rn-Drd1a-C2 no.317031, Rn-Drd2-C1 no.315641 |
| Chemical compound, drug | Biocytin hydrochloride | Sigma-Aldrich | Cat. #:B1758 CAS No:98930-70-2 | |
| Chemical compound, drug | Tetrodotoxin | Latoxan | Cat. #:L8503 CAS No:4368-28-9 | |
| Chemical compound, drug | Picrotoxin | abcam | Cat. #:ab120315 CAS No:124-87-8 | |
| Chemical compound, drug | N-acetyl-cysteine (NAC) | Sigma-Aldrich | Cat. #:A9165 CAS No:616-91-1 | |
| Chemical compound, drug | L-Buthionine-sulfoximine (BSO) | Sigma-Aldrich | Cat. #:B2515 CAS No:83730-53-4 | |
| Chemical compound, drug | L-Glutathione (GSH) | Sigma-Aldrich | Cat. #:G4251 CAS No:70-18-8 | |
| Chemical compound, drug | 4',6-Diamidino-2-phenyl-indol -dihydrochlorid | Sigma-Aldrich | Cat. #:D9542 CAS No: 28718-90-3 | |
| Software, algorithm | GraphPad Prism 9 | GraphPad Software | RRID:SCR_002798 | |
| Software, algorithm | pClamp 10 | Molecular Devices, LLC | RRID:SCR_011323 | |
| Software, algorithm | Mini Analysis Program 6.0.3 | Synaptosoft Inc | RRID:SCR_002184 | |
| Software, algorithm | QuPath | PMID:29203879 | RRID:SCR_018257 | |
| Other | $^1$H-MRS Magnetom 7T/68 cm head | Siemens, Erlangen, Germany | | Human spectroscopy |
| Other | $^1$H-MRS horizontal 9.4T/31 cm | Magnex Scientific, Abingdon, UK | | Rodent spectroscopy |

## Human study: participants

Forty-three men, between 20 and 30 years old, were originally recruited for the study. Informed consent was obtained from all participants, who were debriefed at the end, and experiments were performed in accordance with the declaration of Helsinki and approved by the Cantonal Ethics Committee of Vaud, Switzerland (PB_2016–02019). Valid NuAc $^1$H-MRS and behavioral data were obtained from 22 participants (more experimental details can be found in the Participants section in Supplementary methods of ref *Strasser et al., 2020*). Data analyzed here have not been previously analyzed, but it was collected as part of a larger study previously published (*Berchio et al., 2019*; *Strasser et al., 2020*). Out of the initially recruited 43 participants, data from 16 participants could not be fully collected and included in the analysis due to the following reasons: four participants dropped out voluntarily on the scanning day, four participants reported metal implants only on the scanning day, two participants were not suited for the scanner environment due to their anthropometrics, four participants moved during neuroimaging, and for two participants hardware failed during acquisition. While metabolite data was available for 27 individuals, we excluded behavioral data for 5 out of those (2 due to hardware failure and 3 due to task non-compliance). Thus, our brain-behavior analyses are based on 22 subjects. Out of these, 13 participants were successfully recruited for a second occipital cortex $^1$H-MRS scan to serve as a control.

## Effort-related monetary incentivized force task

Our task, termed monetary incentive force (MIF) task (*Berchio et al., 2019*; *Strasser et al., 2020*), combines aspects from the monetary incentive delay task (*Knutson et al., 2001*) and from effort-based decision-making paradigms (*Salamone et al., 2016*; *Pessiglione et al., 2007*). It requires participants

to exert force on a hand grip dynamometer (TSD121B-MRI, Biopac) (*Figure 1b*) at a threshold corresponding to 50% of each participant's maximum voluntary contraction (MVC). We repeated this procedure three times interspersed by breaks of 3 min each to allow for recovery. The highest out of the three values was used to calibrate the threshold. Participants were comfortably seated in front of a computer screen at 90 cm distance and were instructed to keep the same right upper limb position (i.e. upper arm and forearm at 90° angle and hand extended) whenever using the dynamometer. The dynamometer signal was fed in real time to the computer running MIF task (E-Prime software). The task comprised a total of 80 trials that were run during two halves (H1 and H2; see *Figure 1b*) separated by a 3 min break. Each half contained 2 blocks, each having 20 trials: 5 incentivized trials of each of the different incentives (0.2, 0.5, 1) presented in pseudo-random order, and 5 non-incentivized rest trials occurring after every 3 incentivized trials. To earn the displayed incentives, the participant's 50% MVC threshold had to be reached within 2 s and kept above for another 3 s. The experiment was run under two experimental conditions, with half of the participants performing the task in isolation (i.e. their earnings depended solely on their own performance) while the other half in competition. Success was measured by the number of successful experimental trials, and results were computed separately for the two halves of the experiment (H1 and H2; see *Figure 1b*) for three monetary incentives (i.e. 0.2, 0.5, and 1 CHF). See (*Strasser et al., 2020*) for further details.

## Proton magnetic resonance spectroscopy ($^1$H-MRS) acquisition for participants and data processing

The MR measurements were performed on a Magnetom 7 T/68 cm head scanner (Siemens, Erlangen, Germany) equipped with a single-channel quadrature transmit and a 32-channel receive coil (Nova Medical Inc, MA, USA). The NuAc region of interest voxel was defined by the third ventricle medially, the subcallosal area inferiorly, and the body of the caudate nucleus and the putamen laterally and superiorly, in line with definitions of NuAc anatomy identifiable on MRIs (*Smith et al., 2002*). See (*Strasser et al., 2020*). The Cramer-Rao lower bound (CRLB) for GSH was 8.2 ± 2.8%.

## Animals

Adult male Wistar rats (Charles Rivers, Saint-German-Nuelle, France) weighing 250–275 g upon arrival were individually housed with ad libitum access to food and water in a 12 light-dark cycle (lights switched on at 7:00 am) and a constant temperature at 22 ± 2°C. Following a week of acclimatization to the animal facilities, rats were handled for 2 min/day for 3 days prior to the start of the experiments, in order to habituate to the experimenters. Rats used for operant conditioning experiments were placed in reversed light-dark cycle (lights-on at 20:00, lights-off at 8:00) with ad libitum food and water and such experiments were performed during the dark phase of the cycle. Experiments were carried out in accordance with the European Union Directive of September 22, 2010 (Directive 2010/63/EU) and approved by the Cantonal Veterinary Authorities (Vaud, Switzerland; authorization VD3443). All efforts were done to minimize animal number and suffering.

## Anxiety classification

All rats were tested in the EPM to phenotype for anxiety, as previously described (*Hollis et al., 2015*). Briefly, the EPM consisted of two open and two closed arms (45×10 cm each) extending from a 10×10 cm central area. Closed arms had 10 cm high walls. Lighting was maintained at 16–17 lx in the open arms, 10–11 lx in the central area, and 5–7 lx in the closed arms. Rats were placed on the central area, facing a closed arm and were allowed to explore freely for 5 min. Every animal was recorded, and tracking was performed using Ethovision software (Noldus, Wageningen, the Netherlands). Percentage of time in the open arm was calculated using Ethovision (Noldus).

## Saccharine preference test

Animals' saccharine preference was measured using a two-bottle choice paradigm in the home cage. Briefly, animals were habituated to drink from two water bottles in the home cage. At the start of the saccharine preference test, one water bottle was replaced with a 0.03% saccharine solution. Animals were given free access to both bottles for a period of 48 hr, during which consumption was measured twice daily during light and dark cycles. The location of the saccharine was counterbalanced across groups and switched after 24 hr of consumption to control for any side preference. The percentage

of saccharine solution consumption out of total liquid consumption was analyzed as saccharine preference.

## Light dark box

Anxiety-like behavior in NAC and Veh rats was measured using the Light Dark Box paradigm. The equipment in this test consists of a light (160 lux) and a dark (20 lux) compartment. Animals were initially placed in the dark compartment and then allowed 5 min of total exploration of the apparatus. The parameters evaluated were the number of entries to the light compartment and the latency to enter in the dark compartment after the initial placement in the light compartment.

## Open field

The OF was used to determine locomotor behavior and time spent in the center as a measure of rat's exploration. The OF apparatus and procedure were previously described (*Herrero et al., 2006*). Briefly, the OF consisted of a black circular arena (1 m in diameter, surrounded by walls 32 cm high). For analysis, the percentage of time in the center and total distance walked were calculated. Animals were placed in the center of the arena, and their behavior was monitored for 10 min using a video camera that was mounted from the ceiling above the center of the arena. The light was adjusted to a level of 8–10 lx in the center of the arena. Total distance walked and percentage time in the center were calculated using Ethovision (Noldus).

## Novel object

Immediately after the OF test, rats were exposed to an NO. For this purpose, a small plastic bottle was placed into the center of the OF while the rat was inside. Rats were then given 5 min to freely explore the NO. The time spent exploring the NO was recorded using Ethovision and plotted as a percentage of time with the object.

## Proton magnetic resonance spectroscopy ($^1$H-MRS) for rats

After anxiety and locomotion characterization, rats were transferred to the Center for Bioimaging, École Polytechnique Fédérale deLausanne, where the *in vivo* $^1$H-MRS measurements of brain metabolites was performed. Briefly, all experiments were performed on a horizontal 9.4T/31 cm bore animal MR scanner (Magnex Scientific, Abingdon, UK) using a homemade quadrature $^1$H-coil. Animals were anesthetized and placed in the scanner. Acquisition was done using the spin echo full intensity acquired localized (SPECIAL) sequence in the volume of interest placed in the bilateral NuAc (voxel size: 2×6×3 mm$^3$, TE/TR: 2.8/4000 ms) after acquisition of a set of anatomical T$_2$-weighted images for localization. Field homogeneity was adjusted using FAST(EST)MAP to reach a typical water linewidth of 13.2 Hz in NuAc. Spectra were acquired with 10 blocks of 25 averages, leading to a scan duration of around 30 min, respectively. After post-processing of the spectra, metabolite concentrations as well as the CRLB were determined with LCModel using water as internal reference (*Cherix et al., 2020*).

## Forced swim test

The FST was performed for 15 min under light conditions of 60 lx. Rats were individually placed into glass cylinders (20 cm diameter, 46 cm depth) containing water at 23–25°C. The cylinder was half-filled, in order to prevent the rat from touching the bottom of the cylinder with its feet or tail. The water was changed after each animal's session and the cylinder cleaned. The percentage of immobility time was scored manually using the Observer software (Noldus) by an experimenter blind to each animal's allocation to a group.

## NAC treatment

For experiments including NAC (Sigma Aldrich, A9165) treatment, rats were treated with 500 mg/L in the drinking water, for at least 2 weeks before the beginning of behavioral experiments. Water bottles were changed twice per week to ensure NAC integrity. For operant experiments, treatment was provided starting on the last FR training day and continued until the end of experiments.

## Stereotactic surgery for cannula implantation

Rats were anesthetized by isoflurane inhalation (4%, for 4 min) in an induction chamber and maintained afterward with 2% isoflurane with a flow of 4 l/min. Stereotactic surgery was performed as

previously described (*Hollis et al., 2015*; *van der Kooij et al., 2018a*; *van der Kooij et al., 2018b*). Briefly, rats were mounted on a stereotactic frame (Kopf Instruments, Tujunga, CA, USA), an incision was made along the midline of the skull, the periosteum was removed, and small holes were drilled for the implantation of guide cannulae (Invivo1, Roanoke, VA, USA). Coordinates for NuAc were taken from the Paxinos and Watson brain atlas, relative to bregma, as follows: anterior-posterior: +1.2, mediolateral: ±1.5, and dorsoventral: –6.50. Cannulae were fixated on the skull with three anchoring screws and Paladur acrylic dental cement (Kulzer, Hanau, Germany). Correct cannula placement was confirmed in the end of experiments.

## BSO infusions

Behavioral experiments were performed 24 hr after BSO (Sigma Aldrich, B2515) or Sal administration. Groups were randomized. BSO was dissolved in Sal (AppliChem, A3006,05) at a concentration of 7 µg/µl. For local intra-cerebral infusions, the dummy cannulae were removed, and injectors were inserted extending 2 mm from the guide cannulae. Drugs were bilaterally infused intra-NuAc at a volume of 1 µl per hemisphere at a rate of 0.3 µl/min. The injector remained in place for one additional minute after infusion to allow proper diffusion.

## Progressive ratio task

After at least 10 days after surgery or after introduction to the reversed light-dark cycle, animals started training in an FR1 reinforcement schedule. Operant chambers (Coulbourn Instruments, Holliston, MA, USA), placed in sound attenuating cubicles, were equipped with a grid, underneath which a tray with standard bedding material was placed for collection of feces and urine after each training session. Each chamber had one food tray and two ports placed on either side of the tray. A cue light was placed in each port and the food tray, whereas a house light was placed above the food tray. The right-hand side port of each chamber was designated as 'active', meaning that spontaneous nosepoking would result in the drop of one 45 mg food pellet (Bio-Serv, Flemington, NJ, USA) to the food tray. Upon nosepoking in the active port, the cue and house lights turned off, while the tray light turned on and the pellet dropped to the food tray. The two ports remained inactive for 20 s, during which nosepokes would not result in the delivery of a new pellet (time-out period). Subsequently, the chamber returned to its initial condition. Each training session lasted maximally 2 hr or until a rat acquired 100 pellets. Each rat received six training sessions (one training on each day for five consecutive days, followed by 2 days without training and one more training session on day 8). Only rats that finished at least two training sessions acquiring 100 pellets before the 2-hr mark were used for PR reinforcement schedule (PR test) experiments. To test motivated behavior, rats were exposed to a PR test. PR test sessions were identical to training sessions except that the operant requirement in each trial (T) was the integer (rounded down) of the function $1.4^{(T-1)}$ starting at one nosepoke for the first three trials and increasing in subsequent trials, as has been previously described (*Wanat et al., 2013*). Correct nosepokes (i.e. nosepokes in the active port and outside the timeout period, thus resulting in food delivery), number of obtained sucrose pellets (rewards), and the last ratio completed (break-point) were calculated to evaluate behavioral performance (*Soares-Cunha et al., 2016*; *Richardson and Roberts, 1996*).

## Modified reward learning paradigm in operant boxes

For the modified reward learning paradigm, operant chambers featured two levers on either side of the food tray. The right-hand side lever was designated as 'active', meaning that a lever press resulted in the delivery of a food pellet to the food tray. A house light placed above the food tray was on during the whole training session. An acoustic cue (4.5 kHz, 75.6 dB tone) was provided to initiate the active period, during which a correct lever press led to activation of the tray light and delivery of the pellet. The two levers remained inactive for 20 s and did not deliver a new pellet if pressed during this time-out period. Subsequently, the chamber returned to its initial condition. Each training session lasted maximally 2 hr or until a rat acquired 100 pellets. Each rat received seven training sessions (one training on each day for four consecutive days, followed by 2 days without training and three more training session on days 7 and 8). BSO or Sal was infused 24 h prior to the beginning of the first training session. Two rats from the Sal group were excluded because they did not meet the criterion (100 rewards in less than 2 hr of session in at least two sessions).

## HPLC analysis of GSH levels in brain samples

Animals were decapitated, and their brains were quickly removed, frozen in isopentane on dry ice at a temperature between −50 and −40°C, and stored at −80°C until further processing. Coronal sections (200 µm thick) were punched to obtain the brain tissue of NuAc region as previously described (*Guitart et al., 2000*). For GSH measurements, brain samples were briefly sonicated in Eppendorf vials containing 100 µl of 0.5 M perchloric acid +100 µM EDTA-2Na and centrifuged at 16,000 $g$ for 10 min at 4°C. The supernatant was collected, filtered through 0.22 µm filters (5000 g for 30″ at 4°C) and used for HPLC analysis. Levels of GSH were assessed by reverse-phase HPLC with electrochemical detection (HPLC-ECD stand-alone system, HTEC-500). Using a mobile phase, consisting of 27.59 g/l sodium phosphate monobasic monohydrate (pH2.5), 20 mL/l methanol, 10 mg/l EDTA-2Na, and 100 mg/l SOS dissolved in Milli-Q water, the GSH was separated in a reversed-phase separation column EICOMPACK SC-3ODS using a Gold electrode (ref WE-AU).

## Ex vivo electrophysiology

Rats were anesthetized with isoflurane and decapitated. The brain was quickly removed, and coronal slices (250 µm thick) containing the ventral striatum were cut using a vibrating tissue slicer (Campden Instruments) in oxygenated (95% $O_2$/5% $CO_2$) ice-cold modified artificial CSF (ACSF), containing (in mM): 105 sucrose, 65 NaCl, 25 NaHCO$_3$, 2.5 KCl, 1.25 NaH$_2$PO$_4$, 7 MgCl$_2$, 0.5 CaCl$_2$, 25 glucose, 1.7 L(+)-ascorbic acid. Slices recovered for 1 hr at 35°C in standard ACSF containing (in mM): 130 NaCl, 25 NaHCO$_3$, 2.5 KCl, 1.25 NaH$_2$PO$_4$, 1.2 MgCl$_2$, 2 CaCl$_2$, 18 glucose, 1.7 L(+)-ascorbic acid, and complemented with 2 Na-pyruvate and 3 myo-inositol. In the recording chamber, slices were superfused with oxygenated standard ACSF. MSNs in the NuAc core were patched in the whole-cell configuration with borosilicate pipettes (3–4 MΩ) filled with a KGluconate- or CsGluconate intracellular solution complemented with 0.1% biocytin.

Measurements of intrinsic excitability were conducted at nearly physiological temperature (30–32°C), with an intracellular solution containing (in mM): 130 KGluconate, 10 KCl, 10 HEPES, 10 phosphocreatine, 0.2 EGTA, 4 Mg-ATP, 0.2 Na-GTP (290–300 mOsm, pH 7.2–7.3). To elicit neuronal firing, MSNs were held at −70 mV with direct current injections in the current clamp configuration, using bridge compensation, and depolarization was provided by 5 s long current ramps of increasing magnitude (maximal current ranging from 50 to 300 pA). Recordings were conducted during the first 5 min after establishment of the whole-cell condition. The rheobase (minimal current required to elicit spiking) and the firing threshold were measured as the level of current and voltage, respectively, that induced the first action potential in the ramp protocol. Input resistance (R$_i$) and cell capacitance (C$_m$) were evaluated from the passive response to a −10 mV hyperpolarizing step provided from a holding potential of −60 mV.

mEPSCs were recorded in MSNs voltage-clamped at −60 mV using pipettes (3–4 MΩ) filled with (in mM): 120 CsGluconate, 10 CsCl, 10 HEPES, 10 phosphocreatine, 5 EGTA, 4 Mg-ATP (290–300 mOsm, pH 7.2–7.3). Recordings were conducted at room temperature in the presence of the GABA$_A$R blocker picrotoxin (0.1 mM) and the Na$^+$ channel blocker tetrodotoxin (0.001 mM). Synaptic currents were acquired for 5 min starting from >8 min after the establishment of the whole-cell configuration, to ensure proper diffusion of the intracellular solution.

At the end of each recording, the patch pipette was gently retracted from the cell body to cause membrane re-sealing. Slices were fixated in 4% paraformaldehyde (PFA) overnight, then stored in phosphate-buffered saline (PBS) complemented with 30% sucrose until processing.

Electrophysiological data were acquired through a Digidata1550A digitizer. Signals were amplified through a Multiclamp700B amplifier (Molecular Devices), sampled at 20 kHz, and filtered at 10 kHz using Clampex10 (Molecular Devices). Data were analyzed using Clampfit10 (Molecular Devices). For detection of mEPSCs, traces were filtered at 1 kHz and analyzed using the MiniAnalysis Program (Synaptosoft Inc, Decatur, USA), setting a detection threshold of twice the root mean square of the noise level. Detected events were verified by visual inspection. To construct cumulative frequency plots, the first 200 events recorded in each cell were considered.

## Post-hoc identification of D1- and D2-MSNs

Post-hoc in situ hybridization (RNAscope Multiplex Fluorescent Reagent Kit v2. Assay, Advanced Cell Diagnostics, Inc) for *Drd1* and *Drd2* was performed after patch-clamp electrophysiology following

the manufacturer's instructions, with some modifications. In order to visualize biocytin-filled neurons, brain slices were washed (5 min at RT × three times) in PBS and incubated (overnight at 4°C) with streptavidin-Alexa 488 (ThermoFisher Scientific, S-11223, 1:500) in PBS-triton 0.05%. In the following day, slices were washed (5 min at RT × three times) in PBS, mounted in Superfrost Plus Adhesion Microscope Slides (Thermo Scientific), baked at 60°C for 30 min, and post-fixed in pre-chilled 4% PFA in PBS (15 min at 4°C). Afterward, slices were dehydrated (50% EtOH for 5 min, 70% EtOH for 5 min, twice with 100% EtOH for 5 min at RT) and air dried (5 min). Endogenous peroxidase activity was blocked by RNAscope $H_2O_2$ treatment (10 min at RT), and sections were permeabilized by RNAscope Target Retrieval (5 min at 60°C) and a RNAscope Protease III (30 min at 40°C). Next, slices were hybridized with pre-warmed probe pairs to target *Drd1* and *Drd2* mRNA (channel C1, Rn-Drd2-C1 no.315641) by incubation for 2 hr at 40°C. Signal was amplified by incubation with RNAscope Multiplex FL v2 AMP1 (30 min at 40°C), RNAscope Multiplex FL v2 AMP2 (30 min at 40°C), and RNAscope Multiplex FL v2 AMP3 (15 min at 40°C). Channel C1 signals were developed by incubation with RNAscope Multiplex FL v2 HRP-C1, followed by incubation with Opal 570 (1:1500 diluted in RNAscope Multiplex TSA Buffer), and finally RNAscope Multiplex FL v2 HRP blocker. The same procedure was used to target *Drd1* mRNA (channel C2, Rn-Drd1a-C2 no.317031-C2). The channel C2 signals were developed by incubation with RNAscope Multiplex FL v2 HRP-C2, followed by incubation with Opal 690.

Confocal images were acquired on a LSM 700 confocal microscope (Carl Zeiss) with ×20/0.8 NA air and ×40/1.3 NA oil-immersion objectives (Bioimaging and Optics Platform, BIOP, EPFL). Biocytin-positive cells (113 successfully recovered out of 141 recorded cells) were identified as D1-MSNs (61 out of 113) or D2-MSNs (49 out of 113) based on reactivity for Drd1 and Drd2 RNA. One single neuron was D1-D2 double-positive, and other two neurons were D1-D2 double-negative.

## NeuN and caspase 3 immunohistochemistry

For NeuN and cleaved caspase 3 analysis, free-floating coronal sections (10 or 30 μm thick) were mounted and fixed for 30 min with 4% PFA in PBS at RT, and subsequently rinsed with PBS-containing 0.3% Triton-X-100 (PBS-T), followed by a blocking step of 1 hr incubation in PBS containing 3% BSA. Primary antibodies, anti-NeuN (Millipore, 1:1000) or anti-cleaved caspase 3 (Cell Signaling, 1:200), were diluted in blocking solution and sections incubated overnight at 4°C with gentle shaking. Sections were then washed three times in PBS and incubated with secondary antibodies (anti-mouse Alexa Fluor 488, 1:1000, or anti-rabbit Alexa Fluor 488, 1:1000) for 2 hr at room temperature. After three rinses in PBS, sections were incubated for 5 min with 4,6-diamidino-2-phenylindole (Sigma, D9542), washed again three times in PBS, and then mounted in Fluoromount-G (SouthernBiotech, 0100–01). NeuN and cleaved caspase 3 expression were assessed in the NuAc using an LSM 710 laser-scanning confocal microscope (Carl Zeiss) imaged using a ×10 with a 1.0 digital zoom. Images were analyzed for number of NeuN- or caspase-3-activated positive cells using QuPath positive cell detection (University of Edinburg). At least two sections from each animal were measured and averaged to generate one value per hemisphere per animal for each drug infusion.

## Statistical analyses

Detailed parameters from statistical tests are reported in figure legends and in Stats table indicating the statistical test used, sample size ('**N** indicates the number of animals used for each experiment, and the number of observations (**n**) indicated for electrophysiological recordings refers to single cells derived from seven vehicle-treated and six NAC-treated rats'), as well as degree of freedom, F- and p-values.

For participants, data were analyzed with the *jamovi* software v2.2.5, and correction for multiple comparisons was performed in RStudio v1.1.456 running R version 3.6.1 using function *p.adjust*. The Spearman correlation coefficient was used throughout to keep comparisons between correlation coefficients consistent, even if not all variables violate normality assumptions (tested with a Shapiro-Wilk test). The p-values from multiple correlations within reward levels and for post-hoc analyses were corrected for a false discovery rate of 0.05 using the *p.adjust* default Benjamini–Hochberg procedure. To account for the influence of other metabolites in pairwise correlation analysis, significant correlations were followed by a regression analysis, where the relationship between metabolite of interest (MOI) and target performance was controlled by adding regressors corresponding to the remaining metabolites. To avoid collinearity when using the remaining metabolites as regressors, principal

component analysis (PCA) followed by varimax rotation was used to transform the metabolite variables into orthogonal principal components (PCs) without compression. Only PCs where the MOI does not load were used as regressors. This allowed us to control for as much variance as possible without collinear regressors. Metabolites with significant associations were dichotomized into low and high with median split, and performance was modeled with a 2×2×2 mixed design ANCOVA with half (first, last) as the within subjects' factor, reward (low, high), and metabolite level (low, high) and the principal components that do not load each metabolite as in the previous analysis. Significant triple interaction half *reward*metabolite level was followed by post-hoc analyses testing the double interaction half*metabolite level in two similar 2×2 mixed designed ANCOVAs for low and high reward, which were finally followed by two independent samples t-tests to assess the difference between high and low metabolite level for both the first and last half and two paired samples t-tests to assess the changes due to experiment halves within each reward group. Assumptions of normality and homogeneity of variances were tested with the Shapiro-Wilk and Levene's tests, respectively. All tests were two-tailed, except for post-hoc paired sample t-tests where the direction of the effect was known, with a critical probability of $p \leq 0.05$.

Absence of effects was examined using the appropriate Bayesian statistical test with JASP v0.16.3.0 (University of Amsterdam, Amsterdam, the Netherlands) by computing its Bayes factor (BF01) in favor of the null hypothesis (H0) over the alternative hypothesis (H1).

For animal experiments, statistical analyses were performed with Prism software v8.4.3 (GraphPad software, San Diego, CA, USA) or statistical package for social sciences (SPSS) version 17 (SPSS Inc, Chicago, IL, US) using a critical probability of $p \leq 0.05$. All values are represented as mean ± SEM. For two-group comparisons, one- or two-tailed t-test or Mann-Whitney test was applied, as appropriate. When the effect of more than one independent variable was analyzed, two- or three-way ANOVA was used, followed by post-hoc tests, as appropriate (Holm-Sidak post-hoc test). For survival analysis, the Mantel-Cox test was used.

## Acknowledgements

The authors thank Dr. Ting Yin and Dr. Hongxia Lei for expert help with the performance of MRS in rats and Jocelyn Grosse and Isabelle Guillot de Suduiraut for expert help with animals' surgeries and behavioral testing. This project has been supported by grants from the Swiss National Science Foundation [CR20I3-146431, 31003 A-176206 and −197942 to CS; and SNSF-Spark grant (No 196558) to JR, European Union's Horizon 2020 research and innovation programme under the Marie Sklodowska-Curie grant (N° 895562) to ER-F, NCCR Synapsy (51NF40−158776 and 185897)], Société des Produits Nestlé SA; and intramural funding from the EPFL to CS. FH is currently supported by a VISN7 research development award from the VA. The funding sources had no additional role in study design, in the collection, analysis and interpretation of data, in the writing of the report or in the decision to submit the paper for publication. This paper reflects only the authors' views and the European Union is not liable for any use that may be made of the information contained therein. [1]H-MRS experiments were also supported financially by the Center for Biomedical Imaging (CIBM) of the University of Lausanne (UNIL), University of Geneva (UNIGE), Geneva University Hospital (HUG), Lausanne University Hospital (CHUV), Swiss Federal Institute of Technology (EPFL) and the Leenaards and Louis-Jeantet Foundations.

## Additional information

### Competing interests

Laura Trovo: LT is employee from Nestle S.A. Pascal Steiner: PS is employee from Nestle S.A. Nicolas Preitner: NP is employee from Nestle S.A. The other authors declare that no competing interests exist.

## Funding

| Funder | Grant reference number | Author |
|---|---|---|
| Swiss National Science Foundation | CR2013-146431 | Carmen Sandi |
| Horizon 2020 - Research and Innovation Framework Programme | 895562 | Eva Ramos-Fernández |
| Swiss National Science Foundation | 31003 A-176206 | Carmen Sandi |
| Swiss National Science Foundation | 31003 A–197942 | Carmen Sandi |
| Swiss National Science Foundation | Spark 196558 JR | João Rodrigues |
| NCCR Synapsy | 51NF40–158776 | Carmen Sandi |
| NCCR Synapsy | 185897 | Carmen Sandi |

The funders had no role in study design, data collection and interpretation, or the decision to submit the work for publication.

## Author contributions

Ioannis Zalachoras, Data curation, Formal analysis, Investigation, Visualization, Methodology, Writing – original draft, I.Z. performed the behavioral and molecular experiments in rats and analyzed the data; Eva Ramos-Fernández, Data curation, Formal analysis, Writing – review and editing, E.R. performed the behavioral and molecular experiments in rats and analyzed the data; Fiona Hollis, Data curation, Formal analysis, Investigation, Methodology, F.H. performed the behavioral and molecular experiments in rats and analyzed the data; Laura Trovo, Conceptualization, Supervision, Project administration, Writing – review and editing; João Rodrigues, Formal analysis, Visualization, Writing – review and editing, J.R. analyzed the data from the behavioral and MRS experiments in humans; Alina Strasser, Data curation, Formal analysis, A.S. performed the behavioral and MRS experiments in humans; Olivia Zanoletti, Data curation, Formal analysis, Methodology; Pascal Steiner, Conceptualization, Supervision, Funding acquisition, Project administration, Writing – review and editing; Nicolas Preitner, Supervision, Project administration, Writing – review and editing; Lijing Xin, Data curation, Supervision, Methodology, L.X. performed the behavioral and MRS experiments in humans; Simone Astori, Data curation, Formal analysis, Investigation, Visualization, Writing – review and editing, S.A. performed and analyzed the ex vivo electrophysiological recordings; Carmen Sandi, Conceptualization, Supervision, Funding acquisition, Writing – original draft, Project administration, Writing – review and editing

## Author ORCIDs

Ioannis Zalachoras ⓘ http://orcid.org/0000-0002-8113-3512
Eva Ramos-Fernández ⓘ http://orcid.org/0000-0002-3771-2189
Fiona Hollis ⓘ http://orcid.org/0000-0001-6559-5736
Simone Astori ⓘ http://orcid.org/0000-0001-7698-8332
Carmen Sandi ⓘ http://orcid.org/0000-0001-7713-8321

## Ethics

Human subjects: The informed consent, and consent to publish, was obtained. The experiment was performed in accordance with the Declaration of Helsinki and approved by the Cantonal Ethics Committee of Vaud, Switzerland.

All experiments were performed with the approval of the Cantonal Veterinary Authorities (Vaud, Switzerland) and carried out in accordance with the European Union Directive of 22nd September 2010 (Directive 2010/63/EU). All surgery was performed under sodium pentobarbital anesthesia, and every effort was made to minimize suffering.

## Decision letter and Author response

Decision letter https://doi.org/10.7554/eLife.77791.sa1
Author response https://doi.org/10.7554/eLife.77791.sa2

# Additional files

## Supplementary files
• Transparent reporting form

## Data availability
All data generated or analysed during this study are included in the manuscript and supporting files; Source Data files have been provided for all the figures.

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
