## [Editor Report]

This study demonstrates that the level of glutathione in the nucleus accumbens correlates with effortful behaviors both in humans and in rats. By manipulating enzymes involved in the synthesis of glutathione, the authors provide convincing evidence for the causal involvement of glutathione in this process. The results are surprising and provide an important new dimension by which effortful behaviors are regulated.

---

## [Decision Letter]

**Decision letter after peer review:**

Thank you for submitting your article "Glutathione in the nucleus accumbens regulates motivation to exert reward- incentivized effort" for consideration by *eLife*. Your article has been reviewed by 3 peer reviewers, including Naoshige Uchida as the Reviewing Editor and Reviewer #1, and the evaluation has been overseen by Kate Wassum as the Senior Editor. The following individual involved in review of your submission has agreed to reveal their identity: Hanneke EM den Ouden (Reviewer #2).

Essential revisions:

This study use both humans and rats to demonstrate that the level of glutathione in the nucleus accumbens correlates with effortful behaviors. The authors provide causal evidence for glutathione in rats by manipulating enzymes involved in the synthesis of glutathione. Although how exactly glutathione regulates effort-related behavior remains to be clarified, this study provides intriguing observations and causal evidence. As summarized below all the reviewers found the manuscript to be potentially very interesting. However, they identified some issues that need to be addressed before publication in *eLife*.

1) Even when multiple statistical tests were performed in a given component of the study, the authors have not corrected the significance level for multiple comparisons. This should be corrected. Furthermore, at various points, analyses were performed on subgroups / subsets of the data, and differences in terms of statistical significance were interpreted as a significant difference between the groups. Instead, these differences between groups/subsets should be tested directly. Only if significant, post-hoc simple tests are justified. Please refer to the reviewer 2's comments for further suggestions and specific points.

2) The authors conclude that the GSH level in the nucleus accumbens is critical for effortful behaviors but the authors do not examine other brain regions both in humans and in rats. This makes it difficult to distinguish whether the correlation observed in humans is specific to this brain region or a brain-wide phenomenon. Although the causal experiments in rats partially address this issue, it would be very helpful to know whether the GSH level co-fluctuates widely in the brain or is altered specifically in the nucleus accumbens. If 1H-MRS data has already been acquired, it would be very useful to include in the manuscript. If the specificity of nucleus accumbens is difficult to establish, conclusions would need to be adjusted.

3) Conversely, whether various manipulations in the rat nucleus accumbens specifically affected effortful behaviors or compromised the function of the nucleus accumbens generally is unclear. Some results based on other behaviors would be very helpful to distinguish these possibilities. Furthermore, it would be helpful to discuss (further) how the level of GSH levels can affect effort specifically.

4) It is important to distinguish whether the results in the causal experiments in rats are due to a specific effect of various manipulations on the GSH levels or some general loss of function (e.g. toxicity) in cells in the nucleus accumbens. To address this point, the authors examined whether the BSO effects were reversible but the results in Figure 4 are not convincing. The difference between the groups appears to be mainly due to a reduced statistical power in the BSO group. A better dataset or analyses are required to make this result convincing.

5) Please address the limitation of the exclusion of females from this study.

We hope that the authors can address the above essential points by additional data (and potentially additional experiments) and analyses. Below the reviewers have also provided more detailed comments and additional points to which we would like to see your response. We hope that you can address these issues within 2-3 months.

*Reviewer #1 (Recommendations for the authors):*

1. It is unclear how the level of glutathione regulates effortful behaviors. Glutathione is synthesized with glutamate, glycine and cysteine. Is it possible that some alterations in the glutathione synthesis pathway, either naturally occurring or experimentally, causes some effects on neurotransmitters such as glutamate or glycine. It would be useful to discuss possibilities of such indirect effects.

2. Many results are presented after binary classification of high versus low GSH levels. It would be more useful to use correlations as the level of GSH is continuous. At least, it would be good to increase scatter plots as in Figure 1c.

3. Figure 2c needs to be a better quality.

4. The authors use the acronym, NuAc, for the nucleus accumbens, but in some places "NAc" is used (e.g. page 9, line 8). Please be consistent.

5. The metabolic pathways as well as enzymes are difficult to understand to those who are not familiar with them. Also, there appear to be different names or acronyms depending on literature. It would be helpful for people outside the field if the authors provide a schematic of metabolic pathways and enzymes as a supplementary figure.

*Reviewer #2 (Recommendations for the authors):*

Construct validity

In the monetary incentivized effort task, the required effort is held constant, while the incentives are varied. However, unless effort required and reward that can be obtained are varied independently (see e.g. the task used in Draper et al. 2018 Neuropsychopharmacology) one cannot attribute the effects to one or the other. In relation to this, why do the authors use the number of successfully completed trials rather than the actual effort exerted? This should be a more sensitive measure, as the definition of a successful trial is effectively a thresholded approximation of effort executed (i.e. more or less than the required 50%).

Generalisability / specificity of the results

This study is performed only in males. Why? This needs to be justified in the paper (I am not convinced there is a good justification). In addition, the MRS results are based on a voxel only in the nucleus accumbens. Therefore, one cannot claim specificity of these findings to the accumbens, and may reflect the metabolic state of the entire brain. This should be acknowledged, and conclusions worded so that specificity is not claimed. This also holds for the animal work, where all manipulations and measures are only done in accumbens.

Statistical analyses

Overall, within all different components of the study, multiple statistical tests are performed. These should be corrected for multiple comparisons. Furthermore, at various points analyses are performed on subgroups / subsets of the data, and differences in terms of statistical significance are interpreted as a significant difference between the groups. Instead, these differences between groups/subsets should be tested directly. Only if significant, post-hoc simple tests are justified. Below I will highlight a few examples of this.

The authors perform 2 x 10 pairwise correlation analysis to assess (one set for all motivation levels, once set for the high reward only). First, statistical inference should be corrected for multiple comparisons (so α level = 0.005). However, as the various MRS measures are likely to be correlated, so in order to attribute effects that are selective for GSH, all MRS measures should be entered as covariates in a single regression, to assess the unique contribution of GSH to performance.

When assessing whether effects are different for different conditions, e.g. for early versus late blocks, or dependent on the incentive magnitude, these should be included as factors in the analysis (e.g. a repeated measures ANOVA with the conditions block x incentive value, and the MRS measures as covariates, or using a mixed model approach). Then only when significant interactions are detected (e.g. incentive magnitude x GSH), can you test post hoc the simple effects of GSH in each condition.

If the effects for the human data remain significant using the correct analyses (particularly including all MRS measures in a single multiple regression model), then these analyses are certainly interesting to include in the current study. However, given the effect size I am not sure that they will, and then I am not sure if they are of much added value to the much more intricate animal work.

*Reviewer #3 (Recommendations for the authors):*

In general the experiments are well designed and follow a linear logic. I have several points related to the experimental design that limit interpretation of the results in the manuscripts current form.

1. The rationale for why these experiments are only performed in males in the introduction is not well justified.

2. The reversibility experiments outlined in Figure 4 are not convincing. The authors find that following a washout period of the BSO compound in a subset of male rats that the rats no longer show statistically significant changes in motivated behavior. Based on these observations, the authors conclude: "These set of data supports the view that lowering accumbal GSH levels leads to a transient impairment in effort-based incentivized performance without inducing toxicity". This is not an accurate assessment because the magnitude of the effect following 'washout' in the subset of rats in identical in strength and direction to the results obtained prior to the washout. The only difference is that the statistical power is reduced because only a subset of rats in investigated and this reduced power prevents the results from achieving statistical significance. To make this claim, the authors have to compare the performance of the same rats before and after the washout period.

3. The miniature excitatory postsynaptic potential analysis of MSNs in Figure 6 is a nice addition, but is incomplete. The effects of N-acetyl-cysteine are investigated in context of changes in motivated behavior, but the authors do not assess how motivated behavior correlates with these changes within subjects. The authors performed elevated plus maze an open field prior to their recordings, but did not perform the motivated behavioral analysis which would have strengthened their findings. This lack of within subjects analysis also make it difficult to assess which of the observed changes are likely to contribute to the observed behavior effects. Finally, the rationale for performing these experiments in the core region is not clear. The review cited by the authors that is used for justification more strongly implicates the shell region in sustained efforts associated with increased motivation.

[Editors' note: further revisions were suggested prior to acceptance, as described below.]

Thank you for resubmitting your work entitled "Glutathione in the nucleus accumbens regulates the motivation to exert reward-incentivized effort" for further consideration by *eLife*. Your revised article has been evaluated by Kate Wassum (Senior Editor) and a Reviewing Editor.

The manuscript has been improved but there are some remaining issues that need to be addressed, as outlined below:

As you will see below, Reviewer 2 requests that the strengths of concluding an absence of effect should be examined using a Bayesian statistical test. Although such a test may not yet be common in some fields, it is rapidly becoming common in human cognitive neuroscience, and of particular importance when interpreting null results. For your information, the reviewer pointed out that an open-source statistics program, JASP (https://jasp-stats.org/), could be useful for this purpose. We would like to see your response to this as well as other issues (making the figure of metabolic pathways into a main figure) before making a final decision.

*Reviewer #1 (Recommendations for the authors):*

The authors have addressed my previous concerns adequately. This study provides a novel mechanism that regulates motivated behaviors and thus will be of interest to those who study neural mechanisms underlying motivated behaviors.

*Reviewer #2 (Recommendations for the authors):*

Overall, the authors did a thorough job in this revision.

I only have a few more suggestions:

Most importantly, for any control analyses where the absence of an effect is concluded, conduct Bayesian statistical tests and provide evidence in favour of the null hypothesis. This is particularly important in the relatively small samples used here. When evidence in favour of the null hypothesis is not convincing, please rephrase the associated conclusions, that you cannot conclude that there was no difference. This is particularly for the null results of the BSO and occipital voxel analyses but holds for all control analyses where the absence of an effect is important.

Great that you also have an occipital voxel in a subset of the participants. To make this transparent (also the fact that this is a very small group), please add the panels C-F in figure 2 also for the occipital voxels (so exactly the same figure). This will highlight the degree of specificity of the finding.

What false discovery rate did you use for the BH procedure? Normally this is done a priori (see e.g. here: https://www.statology.org/benjamini-hochberg-procedure/). How did you compute the corrected p-values?

I really like the added figure of the metabolic pathways. I think it would be helpful for readers to add this to the main manuscript, perhaps in figure 1.

*Reviewer #3 (Recommendations for the authors):*

The authors have adequately addressed my previous concerns and the additional data significantly strengthens the conclusions of the study.

---

## [Author Response]

Essential revisions:This study use both humans and rats to demonstrate that the level of glutathione in the nucleus accumbens correlates with effortful behaviors. The authors provide causal evidence for glutathione in rats by manipulating enzymes involved in the synthesis of glutathione. Although how exactly glutathione regulates effort-related behavior remains to be clarified, this study provides intriguing observations and causal evidence. As summarized below all the reviewers found the manuscript to be potentially very interesting. However, they identified some issues that need to be addressed before publication in eLife.1) Even when multiple statistical tests were performed in a given component of the study, the authors have not corrected the significance level for multiple comparisons. This should be corrected. Furthermore, at various points, analyses were performed on subgroups / subsets of the data, and differences in terms of statistical significance were interpreted as a significant difference between the groups. Instead, these differences between groups/subsets should be tested directly. Only if significant, post-hoc simple tests are justified. Please refer to the reviewer 2's comments for further suggestions and specific points.

Thank you for highlighting these issues and suggesting corresponding changes to the study. We have now addressed them thoroughly by correcting multiple comparisons due to the various correlations between metabolites and performance for false discovery rate. We have also reduced the number of performed correlations by merging two pairs of metabolites that share common pathways and by excluding data from one metabolite that had excessive missing values. We have also assessed triple interactions in full factorial models and performed post-hoc tests (with significance levels corrected for multiple comparisons) only after significant interactions.

For more information on the steps undertaken, we would like to refer to our detailed responses to Reviewer 2’s comments below.

2) The authors conclude that the GSH level in the nucleus accumbens is critical for effortful behaviors but the authors do not examine other brain regions both in humans and in rats. This makes it difficult to distinguish whether the correlation observed in humans is specific to this brain region or a brain-wide phenomenon. Although the causal experiments in rats partially address this issue, it would be very helpful to know whether the GSH level co-fluctuates widely in the brain or is altered specifically in the nucleus accumbens. If 1H-MRS data has already been acquired, it would be very useful to include in the manuscript. If the specificity of nucleus accumbens is difficult to establish, conclusions would need to be adjusted.

This is an important point that we could address by analyzing data on GSH levels in other regions that were acquired as well during the 1^H^-MRS scans (mPFC in rats and OL in humans) and by measuring GSH by HPLC (in mPFC of rats). GSH levels were not different in the OL of humans and mPFC of rats categorized by median NuAc GSH levels, neither in the cohort used for the 1^H^-MRS experiments, nor for the cohort used for operant experiments and measured by HPLC. We have included these data in Figure1—figure supplement 1d, for the human data, in Figure 2—figure supplement 1a-c (measured by 1^H^-MRS) and Figure 3-supplement figure 1a (measured by HPLC), for the rats. In the Figure 2—figure supplement 1a-c, the left panel shows the plot with the GSH levels in L/H-GSH levels (median-split by GSH levels in the NuAc); the right panel shows the ROI corresponding to the mPFC in the coronal and sagittal sections during the scanning process. In Figure 3- supplement figure 1a we added the GSH levels in mPFC expressed by % of change vs L-GSH group.

3) Conversely, whether various manipulations in the rat nucleus accumbens specifically affected effortful behaviors or compromised the function of the nucleus accumbens generally is unclear. Some results based on other behaviors would be very helpful to distinguish these possibilities. Furthermore, it would be helpful to discuss (further) how the level of GSH levels can affect effort specifically.

We have addressed this point in several ways.

First, given the NuAc’s involvement in reward processing and anhedonia (Berridge and Kringelbach, 2008), we have included data on saccharine preference in H- and L-GSH rats. As shown in the new panel *n* in Figure 3—figure supplement1, we found no significant differences in sucrose preference between these two groups. This is in addition to the lack of differences between groups that we had already reported in anxiety-like, exploratory, and appetitive behavior (Panels b, c, d, e, and m in Figure 3 Supplement1). Therefore, a NuAc non-effortful behavior, such as the hedonic behavior inherent to saccharine preference is not related to current findings regarding GSH-related differences in NuAc.

Second, we performed additional experiments to exclude that the observed BSO effects were due to non-specific actions that could have compromised NuAc integrity or function, and have conducted several subsequent analyses. At the behavioral level, we tested animals in tasks that engage the NuAc (Berridge and Kringelbach, 2008) but do not require effortful behaviors, such as consumption of palatable rewards and operant learning at a low fixed ratio (FR1). When given free access to 20 sucrose pellets (accounting for the maximum they get access in a progressive ratio session) 24h following either Sal or BSO intra-NuAc infusions (i.e., a time point that corresponds with the timing when animals were tested in the progressive ratio following these treatments), animals from both groups did not differ as they all ingested all the pellets within 15 minutes (Figure 4—figure supplements 1j). Then, to assess performance in a FR1 task, we divided animals previously trained and tested in the standard progressive ratio task and formerly infused with Sal into two groups, one of them receiving intra-NuAc Sal and the other BSO. In this case, FR1 training was modified as follows: training was performed in different operant boxes, involving levers instead of nosepokes and a tone as discriminative stimulus (instead of house light). We also changed the environment, including a new floor. No differences were observed between groups (Figure4—figure supplement 1k), indicating that BSO in the NAc does not interfere with incentivized operant learning that requires minimum effortful behavior (i.e., FR instead of PR).

Furthermore, in order to rule out a possible toxic effect on the NuAc of the BSO dose used, we performed an activated caspase 3 (a marker of the execution-phase of cellular apoptosis) and neuN (a marker of mature neurons) staining assays on accumbal sections from a subset of animals tested in the PR test (Figure 4—figure supplement1 m and o). No staining differences were observed between Sal and BSO groups, indicating a lack of BSO toxicity at the dose used in this experiment.

These new experiments and text have now included in the main text and corresponding supplementary figures.

4) It is important to distinguish whether the results in the causal experiments in rats are due to a specific effect of various manipulations on the GSH levels or some general loss of function (e.g. toxicity) in cells in the nucleus accumbens. To address this point, the authors examined whether the BSO effects were reversible but the results in Figure 4 are not convincing. The difference between the groups appears to be mainly due to a reduced statistical power in the BSO group. A better dataset or analyses are required to make this result convincing.

To address this point, we have included a new cohort of animals for the BSO washout experiment, increasing the number of animals per group (from 8 to 17 in both Sal and BSO groups). As can be observed in the new Figure 4g-k, the two groups do not differ in their breakpoint, rewards obtained, or correct nosepoke number and percentage, indicating that the BSO effect in the operant performance is not observed after a washout period (here measured 4 weeks after treatment).

In this revision, we have also increased the number of samples processed for potential toxicity (from 4 reported in our first submission to 10 here, see Figure 4—figure supplement1 m and o) and reanalyzed the images with a more accurate method (QuPath software included in material and methods section) that detects automatically in DAPI positive cells the presence of fluorescence in the green channel (corresponding to activated Casp3 or NeuN). The same results were obtained: lack of toxicity of the BSO treatment used in our study.

5) Please address the limitation of the exclusion of females from this study.

We fully agree and are deeply convinced on the imperative need of including both sexes in any biomedical work. This is currently the practice in our laboratory, but we have been ‘victims’ of the general trend in life sciences for many years. In this case, the study started by evaluating data from a human MRS study that started already 8-10 years ago, and that was originally motivated by former data in the lab on social hierarchies in rodents (and given the sex-specificity of the nature of social competitions, it involved only males). Given that our analyses of the metabolites identified the potential importance of GSH in the NuAc in the context of effortful motivation, we set up experiments in rodents to address the causal importance of this antioxidant. In order to progress mechanistically in this question, we back translated the studies in men to male rats. Our first findings were encouraging and we set up a collaboration to investigate the observed phenomenon more deeply.

But then, the pandemic hit and we had to sacrifice animals, stop experiments and limit our original ambitious greatly. We remain committed to perform studies in the two sexes, both in humans and rodents, and currently have launched one study (that will be followed by a second related one in humans) in which we will have a larger sample of both men and women and study metabolites in different brain regions in the context of motivated behavior. We will perform follow up studies in rodents, in a similar way as in the current study, to address causality.

However, we strongly feel now that the set of data we have managed to gather and complete so far, as reported in this manuscript, is critically new and important and deserves publication as soon as possible. Completing parallel analyses in females would certainly take an additional period of 2-3 years. We hope very much that editors and reviewers would convene with us on the importance of publishing the study as currently stands.

We have addressed this important point in the manuscript (conclusion) by stating “As our studies were performed in male humans and rodents, it is important to note as a limitation to our findings. We are currently expanding our studies to female populations, where we hope to find similar relationships between GSH and motivated behavior.”

Reviewer #1 (Recommendations for the authors):1. It is unclear how the level of glutathione regulates effortful behaviors. Glutathione is synthesized with glutamate, glycine and cysteine. Is it possible that some alterations in the glutathione synthesis pathway, either naturally occurring or experimentally, causes some effects on neurotransmitters such as glutamate or glycine. It would be useful to discuss possibilities of such indirect effects.

As mentioned in the main text, GSH may act as a glutamate reservoir. In addition, NAC can induce activation of the cystine (CySS)/glutamate (see Appendix) antiporter by providing supplementary CySS. This transporter controls extracellular glutamate levels and regulates glutamate release. For example, in models of schizophrenia, NAC shows a reversed increase in extracellular glutamate accompanied by beneficial effects at the behavioral level (*Baker D.A., Madayag A., Kristiansen L.V., Meador-Woodruff J.H., Haroutunian V., Raju I. Contribution of cystine-glutamate antiporters to the psychotomimetic effects of phencyclidine. Neuropsychopharmacology).* Thus, NAC can restore glutamate levels when needed. In parallel, the antioxidant effect of GSH can regulate NMDAR activity. Since high levels of oxidizing agents reduce NMDAR activity through binding to redox-sensitive extracellular sites, the reduction of oxidants by GSH may recapitulate such condition (*Synaptic plasticity impairment and hypofunction of NMDA receptors induced by glutathione deficit: Relevance to schizophrenia. Neuroscience. 2006*). Our study also demonstrates that intrinsic GSH levels in NuAc can promote reward-based motivation in humans and rats, and this can be mediated by GSH as a source of glutamate and cysteine/cystine, acting through this antiporter and regulating glutamate extracellular levels in NuAc or as antioxidant improving NMDAR activity.

2. Many results are presented after binary classification of high versus low GSH levels. It would be more useful to use correlations as the level of GSH is continuous. At least, it would be good to increase scatter plots as in Figure 1c.

We thank the reviewer for this recommendation. As suggested, we performed a correlation between GSH levels and operant condition performance. Interestingly, we observed a significant correlation between the number of rewards obtained and the levels of GSH in the NuAc. This correlation was lost in rats infused with BSO. These results confirm the effect of GSH levels on operant condition performance, since when we consider GSH levels as continuous and not just a mean threshold, the effect of binary classification is maintained. Indeed, when GSH levels are decreased through BSO, the correlation is lost indicating that this effect is specific to GSH levels in NuAc (included in the Figure 3—figure supplement1l and Figure 4—figure supplement1i ).

3. Figure 2c needs to be a better quality.

Following the reviewer’s advice we have added an image with better quality. We have also included better quality images of the Coronal and Saggital sections of the NuAc (see new Figure 2).

4. The authors use the acronym, NuAc, for the nucleus accumbens, but in some places "NAc" is used (e.g. page 9, line 8). Please be consistent.

We apologize for the inconsistency, that has now been corrected throughout the manuscript.

5. The metabolic pathways as well as enzymes are difficult to understand to those who are not familiar with them. Also, there appear to be different names or acronyms depending on literature. It would be helpful for people outside the field if the authors provide a schematic of metabolic pathways and enzymes as a supplementary figure.

We agree with the reviewer that a schematic figure may improve the understanding of the treatments used in the manuscript and the molecular pathways enhanced/disrupted by them. To aid the reader’s comprehension, we have designed a schematic of the main enzymes and players in the GSH synthesis cycle, which is now added in the Appendix.

GSH metabolic pathways. GSH or γ-L-glutamyl-L-cysteinylglycine, is a tripeptide, that can exist in a reduced (GSH) or in an oxidized (GSSG) state. It is synthesized in the cytosol by an ATP-dependent two-step process catalyzed by γ-glutamyl cysteine synthase/ligase (γ-GCS) and GSH synthetase (GSS), being the γ-GCS the rate-limiting enzyme. GSH is freely distributed in the cytosol (up to 10mM concentration) and compartmentalized in the nucleus, endoplasmic reticulum and mitochondria (up to 15%). GSH participates in the neutralization of H2O2 in the cytosol in collaboration with the glutathione peroxidase (GPx) which takes hydrogens from two GSH molecules, converting one molecule of H2O2 into two of H_2_O and one of GSSG. Glutathione reductase (GSR) reduces GSSG back to GSH using NADPH as an electron donor. The main enzyme that initiates the process of GSH degradation is named glutamyl transpeptidase (GGT). This enzyme is localized in the plasma membrane, facing outwards and acting on the extracellular pool of GSH released by the cells. The degradation of GSH by GGTs generates γ-glutamyl-aminoacids and cysteine-glycine dipeptides, which are subjected to the action of membrane-bound peptidase to yield cysteine (Cys) and glycine (Gly). Therefore, the major function of GGT is to recover GSH precursors from extracellular pools which can be lately taken up by cells and used for the new synthesis of GSH. The *N*-acetyl-cysteine (NAC) is a donor of Cys (the rate-limiting substrate) and the Buthionine sulfoximine (BSO) is a specific inhibitor of γ-GCS. In this work the systemic treatment of NAC in the drinking water was used to increase GSH levels in NuAc and the BSO intra-nucleus accumbens infusion to reduce GSH in NuAc.

Reviewer #2 (Recommendations for the authors):Construct validityIn the monetary incentivized effort task, the required effort is held constant, while the incentives are varied. However, unless effort required and reward that can be obtained are varied independently (see e.g. the task used in Draper et al. 2018 Neuropsychopharmacology) one cannot attribute the effects to one or the other. In relation to this, why do the authors use the number of successfully completed trials rather than the actual effort exerted? This should be a more sensitive measure, as the definition of a successful trial is effectively a thresholded approximation of effort executed (i.e. more or less than the required 50%).

Thank you for this insightful suggestion. We agree with the reviewer that the experimental design in Draper et al. (2018) allows for the parametrization of effort and reward sensitivity and to disentangle both effects. In our paradigm, (relative) effort is constant throughout so it is less clear if drops in successful 1 CHF trials in low GSH participants is due to reward sensitivity or effort sensitivity changes alone. Following a later recommendation from the current reviewer (further expanded the corresponding reply), we modeled our successful trials with a 2x2x2 mixed design ANOVA with block (first, last) as the within subjects’ factor, reward (low, high) and GSH level (low, high). The triple interaction block*reward*GSH level effect was significant, but not the interaction effects of block*GSH level or reward*GSH level. If sensitivity to effort would decrease with time in different GSH levels, one would expect to find the bock*GSH level interaction significant as well. The same reasoning applies to the reward*GSH interaction.

Regarding the use of the number of exerted efforts, it indeed should be more sensitive to energy expenditure but we also expect this measure to be more heterogeneous, since it can be associated with different behavioral decisions (failure to achieve initial threshold, failure to remain above the maintenance threshold and not attempting the trial) that can change throughout the experiment as participants decide which strategy is best. Since the experiment was not designed for this purpose, we left out this analysis. However, there are similar effects to those found for the number of successful trials (i.e., triple interaction in the aforementioned mixed design ANOVA and a significant interaction block*GSH level for the 1 CHF reward). These data were added to the Figure1—figure supplement 1 and the analysis’ results were added to the Figure1-Source Data 1.

Generalisability / specificity of the resultsThis study is performed only in males. Why? This needs to be justified in the paper (I am not convinced there is a good justification).

We fully agree and are deeply convinced on the imperative need of including both sexes in any biomedical work. This is currently the practice in our laboratory, but we have been ‘victims’ of the general trend in life sciences for many years. In this case, the study started by evaluating data from a human MRS study that started already 8-10 years ago, and that was originally motivated by former data in the lab on social hierarchies in rodents (and given the sex-specificity of the nature of social competitions, it involved only males). Given that our analyses of the metabolites identified the potential importance of GSH in the NuAc in the context of effortful motivation, we set up experiments in rodents to address the causal importance of this antioxidant. In order to progress mechanistically in this question, we back translated the studies in men to male rats. Our first findings were encouraging and we set up a collaboration to investigate the observed phenomenon more deeply.

But then, the pandemic hit and we had to sacrifice animals, stop experiments and limit our original ambitions greatly. We remain committed to performing studies in the two sexes, both in humans and rodents, and currently have launched one study (that will be followed by a second related one in humans) in which we will have a larger sample of both men and women and study metabolites in different brain regions in the context of motivated behavior. We will perform follow up studies in rodents, in a similar way as in the current study, to address causality.

However, we strongly feel now that the set of data we have managed to gather and complete so far, as reported in this manuscript, is critically new and important and deserves publication as soon as possible. Completing parallel analyses in females would certainly take an additional period of 2-3 years. We hope very much that editors and reviewers would convene with us on the importance of publishing the study as currently stands.

We have addressed this important point in the manuscript (conclusion) by stating “Future studies are warranted to expand the current analyses to females and to investigate the mechanisms whereby GSH levels in the NuAc affect motivated performance.”

In addition, the MRS results are based on a voxel only in the nucleus accumbens. Therefore, one cannot claim specificity of these findings to the accumbens, and may reflect the metabolic state of the entire brain. This should be acknowledged, and conclusions worded so that specificity is not claimed. This also holds for the animal work, where all manipulations and measures are only done in accumbens.

We have addressed this in Figure 1-supplement 1d for the human experiment (1^H^-MRS results), and Figure 2- supplement 1a-c (region specificity measured by 1^H^-MRS) and Figure 3-supplement figure 1a (specificity of NuAc measured by HPLC) for the animal work. For the human data: Occipital cortex 1^H^-MRS was also performed for a subset of participants (GSH low N=6, GSH high N=7) was tested with a 2x2x2 mixed design ANOVA with block (first, last) as the within subjects factor, reward (low, high) and GSH level (low, high) (Figure1-Source Data 1). The triple interaction block*reward*GSH level effect was not significant (p=0.215). This result was added to the manuscript. For the animal work: we have included the ^1^H-MRS and HPLC data from the mPFC, discussed above in our answers to the Essential revisions-question 2.

Statistical analysesOverall, within all different components of the study, multiple statistical tests are performed. These should be corrected for multiple comparisons. Furthermore, at various points analyses are performed on subgroups / subsets of the data, and differences in terms of statistical significance are interpreted as a significant difference between the groups. Instead, these differences between groups/subsets should be tested directly. Only if significant, post-hoc simple tests are justified. Below I will highlight a few examples of this.The authors perform 2 x 10 pairwise correlation analysis to assess (one set for all motivation levels, once set for the high reward only). First, statistical inference should be corrected for multiple comparisons (so α level = 0.005).

Thank you for this comment. In order to reduce the number of comparisons -and given that several metabolites are known to be metabolically related- we have now merged Cr and PCr metabolites into CrPCr and NAA with NAAG into NAANAAG. We also removed the metabolite GPC since it was only identified in 15 participants (and therefore, many subjects had missing values for this metabolite due to low detection). This led to 7 comparisons per reward level (all rewards, 1 CHF, 0.5 CHF and 0.2 CHF), each of these corrected for false discovery rate with the Benjamini–Hochberg (BH) procedure. Both GSH and PE levels correlated significantly with the total number of 1 CHF rewards (respectively: p_corrected = 0.049; p_corrected = 0.049). Figures 1b and Figure1-supplement figure1 are now updated with these new comparisons and significance levels.

However, as the various MRS measures are likely to be correlated, so in order to attribute effects that are selective for GSH, all MRS measures should be entered as covariates in a single regression, to assess the unique contribution of GSH to performance.

Thank you for this interesting comment, but we respectfully have a different view on this issue. In our view, the correlation between these metabolites does not necessarily suggests that these should be regressed out of each pairwise association between metabolites and performance. We expect metabolites to covary due to common pathways and controlling for all other metabolites could hinder our ability to observe any effect, due to removing too much variation due to individual metabolites. Furthermore, the proposed approach would result in highly colinear predictors which is also undesirable in regression models: modeling the data as suggested results in variance inflation factors (VIFs) ranging from 1.27 to 4.65, with 4 of the 7 predictors with VIFs > 2, which can be concerning. Therefore, to still control for the influence of other metabolites and avoid the collinearity problem, we transformed the 7 variables space into a 7 orthogonal principal component space using PCA, followed by varimax rotation to maximize the uniqueness of each principal component. We then used the components that GSH didn’t load into as covariates. This provided 5 of the 7 principal components, explaining a total of 67.9% of the total variance in the dataset, to be used as covariates. A regression model predicting performance 1 CHF with GSH and these 5 principal components as covariates still has GSH as a significant predictor (standardized coefficient = 0.45, p = 0.031) and no colinear predictors (all VIFs <1.1). Ins and CrPCr are also loading to the principal components that were left out as covariates, meaning some of their influence is not being accounted but if ran a partial Spearman correlation between GSH and 1 CHF performance and control for CrPCr and Ins, the correlation is still present (Spearman’s Ro = 0.41, one-tailed-p=0.036); on the other hand, the pairwise correlations between both CrPCr and Ins, and 1 CHF performance remain not significant after controlling for GSH (one-tailed ps>0.425). Following the same approach for PE, since it is also associated with the response variable, we used 6 of the 7 components that didn’t load PE, explaining a total of 84.79% of the total dataset variance. The component left out loads PE exclusively, meaning the influence from all other metabolites can be accounted for. The regression model predicting performance 1 CHF with PE and these 6 principal components as covariates also still has PE as a significant predictor (standardized coefficient = 0.40, p = 0.035) and no colinear predictors (all VIFs <1.1). Corrected both p-values for FDR using the BH procedure results in p=0.035 and p=0.035 for GSH and PE respectively. Changes were added to the Results section and PCA and VIF information to the Figure1-Source Data 1.

When assessing whether effects are different for different conditions, e.g. for early versus late blocks, or dependent on the incentive magnitude, these should be included as factors in the analysis (e.g. a repeated measures ANOVA with the conditions block x incentive value, and the MRS measures as covariates, or using a mixed model approach). Then only when significant interactions are detected (e.g. incentive magnitude x GSH), can you test post hoc the simple effects of GSH in each condition.If the effects for the human data remain significant using the correct analyses (particularly including all MRS measures in a single multiple regression model), then these analyses are certainly interesting to include in the current study. However, given the effect size I am not sure that they will, and then I am not sure if they are of much added value to the much more intricate animal work.

Thank you for this suggestion. We ran these models for both GSH and PE, dichotomized into low and high with median split, since their scores were both correlating with 1 CHF performance. Hence, we modeled successful trials with a 2x2x2 mixed design ANCOVA with block (first, last) as the within subjects’ factor, reward (low, high) and metabolite level (low, high) and the principal components that don’t load each metabolite as in the previous analysis. The triple interaction block*reward*GSH level was significant (p=0.031, partial eta square = 0.275) while the triple interaction block*reward*PE level was not (P = 0.123, partial eta square = 0.162). Hence, we proceeded with post-hoc tests for GSH. We tested the double interaction block*GSH level in two similar mixed designed ANCOVAs for low and high reward. This factor was significant for high but not low reward (respectively: p_corrected = 0.046, partial eta square = 0.301; p = 0.341, partial eta square = 0.061). Finally, high and low GSH groups were significantly different in the last block (p_corrected = 0.040, Cohen’s D = 1.08) but not in the first block (p_corrected = 0.238) due to a decrease in performance in the low GSH group and a maintenance of performance in the high GSH group (respectively Wilcoxon W = 21.0, p = 0.017, corrected p = 0.034, rank biserial correlation = 1.00; p_corrected = 0.553, Cohen’s D = 0.185).

Reviewer #3 (Recommendations for the authors):In general the experiments are well designed and follow a linear logic. I have several points related to the experimental design that limit interpretation of the results in the manuscripts current form.1. The rationale for why these experiments are only performed in males in the introduction is not well justified.

We fully agree and are deeply convinced on the imperative need of including both sexes in any biomedical work. This is currently the practice in our laboratory, but we have been ‘victims’ of the general trend in life sciences for many years. In this case, the study started by evaluating data from a human MRS study that started already 8-10 years ago, and that was originally motivated by former data in the lab on social hierarchies in rodents (and given the sex-specificity of the nature of social competitions, it involved only males). Given that our analyses of the metabolites identified the potential importance of GSH in the NuAc in the context of effortful motivation, we set up experiments in rodents to address the causal importance of this antioxidant. In order to progress mechanistically in this question, we back translated the studies in men to male rats. Our first findings were encouraging and we set up a collaboration to investigate the observed phenomenon more deeply.

But then, the pandemic hit and we had to sacrifice animals, stop experiments and limit our original ambitious greatly. We remain committed to perform studies in the two sexes, both in humans and rodents, and currently have launched one study (that will be followed by a second related one in humans) in which we will have a larger sample of both men and women and study metabolites in different brain regions in the context of motivated behavior. We will perform follow up studies in rodents, in a similar way as in the current study, to address causality.

However, we strongly feel now that the set of data we have managed to gather and complete so far, as reported in this manuscript, is critically new and important and deserves publication as soon as possible. Completing parallel analyses in females would certainly take an additional period of 2-3 years. We hope very much that editors and reviewers would convene with us on the importance of publishing the study as currently stands.

We have addressed this important point in the manuscript (conclusion) by stating “Future studies are warranted to expand the current analyses to females and to investigate the mechanisms whereby GSH levels in the NuAc affect motivated performance.”

2. The reversibility experiments outlined in Figure 4 are not convincing. The authors find that following a washout period of the BSO compound in a subset of male rats that the rats no longer show statistically significant changes in motivated behavior. Based on these observations, the authors conclude: "These set of data supports the view that lowering accumbal GSH levels leads to a transient impairment in effort-based incentivized performance without inducing toxicity". This is not an accurate assessment because the magnitude of the effect following 'washout' in the subset of rats in identical in strength and direction to the results obtained prior to the washout. The only difference is that the statistical power is reduced because only a subset of rats in investigated and this reduced power prevents the results from achieving statistical significance. To make this claim, the authors have to compare the performance of the same rats before and after the washout period.

In order to address this concern, we have performed a new washout experiment, increasing the number of animals per group (to 17 instead of 8) to increase the statistical power of the results and clarify the time effect of the BSO and its toxicity effect. Again, in this new cohort of animals (new Figure 4g-k), no differences were observed in the breakpoint, rewards or correct nosepoke number and percentage. These results demonstrate that the BSO effect can disappear following a washout of 4 weeks. We have also increased the number of samples for the toxicity assay (from 4 to 10, new Figure4—figure supplement1 m and o) and, thus, strengthen our findings that no toxicity is observed after BSO infusions.

3. The miniature excitatory postsynaptic potential analysis of MSNs in Figure 6 is a nice addition, but is incomplete. The effects of N-acetyl-cysteine are investigated in context of changes in motivated behavior, but the authors do not assess how motivated behavior correlates with these changes within subjects. The authors performed elevated plus maze an open field prior to their recordings, but did not perform the motivated behavioral analysis which would have strengthened their findings. This lack of within subjects analysis also make it difficult to assess which of the observed changes are likely to contribute to the observed behavior effects. Finally, the rationale for performing these experiments in the core region is not clear. The review cited by the authors that is used for justification more strongly implicates the shell region in sustained efforts associated with increased motivation.

The primary aim of the ex vivo electrophysiological experiments was to assess the cellular effects of the N-acetyl-cysteine treatment in MSNs. We have performed these experiments in naïve animals, as this approach has allowed us to isolate the impact of this treatment from the plastic changes expected to be induced in MSNs by the operant behavior paradigm. Plasticity of excitatory synapses within the NuAc contributes to cue-reward association (Andrzejewski et al., 2103; Floresco et al., 2001; Di Ciano et al., 2001; Hernandez et al., 2005; Kelley et al., 1997), and direct link between NMDAR-dependent synaptic plasticity within the NuAc core and the development of cue-evoked neural activity during learning has been confirmed in a recent report (Vega-Villar et al., 2019). Further synaptic changes are likely to be induced by the exposure to the progressive ratio task, although, to our knowledge, the specific NuAc division and cell-type involved have not yet been ascertained. Thus, although we agree with the Reviewer that a correlation between electrophysiological findings and behavioral performance would underscore the relevance of the cellular observations, such a correlational approach would require an extensive investigation with a large number of subjects, in order to 1.: obtain sufficient variability in the behavioral performance distribution, such that the ‘behavioral score’ can be correlated with the adequate statistical power with a ‘cellular excitability score’ extracted from the electrophysiological recordings after behavior; 2.: include the investigation of additional conditions to disentangle the direct effect of N-acetyl-cysteine from the learning-induced modifications in the accumbal circuits (e.g., checking the interaction treatment x PR exposure).

To our understanding, there is no demonstration of an exclusive role of the NuAc shell in sustaining the effort during the progressive ratio paradigm. The review by Floresco (2015) emphasizes that the core and shell division of the NuAc can differentially govern the outcome of motivated behavior. Inactivating the core reduces cue-based responding, whereas inactivating the shell can increase the performance towards an irrelevant, non-rewarded, or less preferable outcome. Thus, whereas the core initiates goal-directed behavior towards the reward, the shell inhibits goal-irrelevant behavior. As we now indicate in the results, in our experiments ex vivo, we prioritized the core because on this initiator role, although we agree that a complete description of the cellular effects of NAC should have included both accumbal divisions. However, due to the procedures required for *post hoc* identification of the recorded cells (biocytin filling, fixation, and successful identification after RNAscope processing), we were forced to limit the number of recordings performed in each slice (typically 2-3 cells) and decided to focus on a single accumbal region in order to maximize the chance of collecting sufficient recordings from both D1- and D2-MSNs.

References:

Andrzejewski, M. E., McKee, B. L., Baldwin, A. E., Burns, L., & Hernandez, P.. The clinical relevance of neuroplasticity in corticostriatal networks during operant learning. Neuroscience and biobehavioral reviews, 37(9 Pt A), 2071–2080 (2013).

Floresco SB. The nucleus accumbens: an interface between cognition, emotion, and action. Annu Rev Psychol. Jan 3;66:25-52 (2015).

Floresco, S. B., Blaha, C. D., Yang, C. R. & Phillips, A. G. Dopamine D1 and NMDA receptors mediate potentiation of basolateral amygdala-evoked firing of nucleus accumbens neurons. J. Neurosci. 21, 6370–6376 (2001).

Di Ciano, P., Cardinal, R. N., Cowell, R. A., Little, S. J. & Everitt, B. J. Differential involvement of NMDA, AMPA/kainate, and dopamine receptors in the nucleus accumbens core in the acquisition and performance of pavlovian approach behavior. J. Neurosci. 21, 9471–9477 (2001).

Hernandez, P. J., Andrzejewski, M. E., Sadeghian, K., Panksepp, J. B. & Kelley, A. E. AMPA/kainate, NMDA, and dopamine D1 receptor function in the nucleus accumbens core: a context-limited role in the encoding and consolidation of instrumental memory. Learn. Mem. 12, 285–295 (2005).

Kelley, A. E., Smith-Roe, S. L. & Holahan, M. R. Response-reinforcement learning is dependent on N-methyl-D-aspartate receptor activation in the nucleus accumbens core. Proc. Natl Acad. Sci. USA 94, 12174–12179 (1997).

Vega-Villar, M., Horvitz, J.C., Nicola, S.M.. NMDA receptor-dependent plasticity in the nucleus accumbens connects reward-predictive cues to approach responses. Nat Commun. Sep 27;10(1):4429 (2019).

[Editors' note: further revisions were suggested prior to acceptance, as described below.]

The manuscript has been improved but there are some remaining issues that need to be addressed, as outlined below:As you will see below, Reviewer 2 requests that the strengths of concluding an absence of effect should be examined using a Bayesian statistical test. Although such a test may not yet be common in some fields, it is rapidly becoming common in human cognitive neuroscience, and of particular importance when interpreting null results. For your information, the reviewer pointed out that an open-source statistics program, JASP (https://jasp-stats.org/), could be useful for this purpose. We would like to see your response to this as well as other issues (making the figure of metabolic pathways into a main figure) before making a final decision.

We would like to thank the editor and reviewers for their consideration and appreciation of our efforts to improve the quality of the manuscript. We have now addressed the different requests from Reviewer 2, as requested. First, we have examined the absence of effect using Bayesian statistical test with the suggested JASP programme. We have also added an additional Figure 1—figure supplement 2 to include the occipital voxels and a representative spectra of the same region. Furthermore, we have included the metabolic pathway diagrams in Figure 1, as suggested. None of the additional analyses modify the results and conclusions of our data, but the opposite, they further reinforce them, strengthening the manuscript and study. See our point-by point response below.

Reviewer #2 (Recommendations for the authors):Overall, the authors did a thorough job in this revision.I only have a few more suggestions:Most importantly, for any control analyses where the absence of an effect is concluded, conduct Bayesian statistical tests and provide evidence in favour of the null hypothesis. This is particularly important in the relatively small samples used here. When evidence in favour of the null hypothesis is not convincing, please rephrase the associated conclusions, that you cannot conclude that there was no difference. This is particularly for the null results of the BSO and occipital voxel analyses but holds for all control analyses where the absence of an effect is important.Great that you also have an occipital voxel in a subset of the participants. To make this transparent (also the fact that this is a very small group), please add the panels C-F in figure 2 also for the occipital voxels (so exactly the same figure). This will highlight the degree of specificity of the finding.What false discovery rate did you use for the BH procedure? Normally this is done a priori (see e.g. here: https://www.statology.org/benjamini-hochberg-procedure/). How did you compute the corrected p-values?I really like the added figure of the metabolic pathways. I think it would be helpful for readers to add this to the main manuscript, perhaps in figure 1.

We thank the reviewer for these additional suggestions. Following their input, we have now conducted a Bayesian statistical test for the absence of effect, especially in cases when small samples were used (i.e., not just for the null results of the BSO and the occipital lobe voxel analysis, but more generally). According to standards in the field, when BF01 was greater than 3, it was considered as moderate evidence in favour of the null hypothesis and stated that there was no difference. When BF01 was lower than 3, we rephrased our statements in the manuscript as “*we observe no statistically significant differences*”. The reviewer can also find the result of the Bayesian tests in the corresponding figure legends in the article file and in the Stats GSH manuscript file. None of these analyses modify the results and conclusions of our data.

We have also included a new Figure1—figure supplement 2 in order to show the occipital voxels used for spectroscopy and a representative spectra of the same region. This figure further supports the specificity of our measurements.

Regarding the false discovery rate, we used the *p.adjust* default Benjamini–Hochberg procedure, which follows the method described in Benjamini and Hochberg (1995). Instead of computing a critical value, this function returns adjusted P values, so they can be compared with false discovery rate (*q**) directly. It sorts the P values in ascending order, multiplies them by the number of comparisons and divides them by their sort rank after. For example:

p.adjust(c(0.03, 0.04, 0.01, 0.02), method = ‘BH’) sorts the (N=4) P values: 0.01, 0.02, 0.03, 0.04 (rank 1, 2, 3, 4).Then adjusts them:0.01*4/1, 0.02*4/2, 0.03*4/3, 0.04*4/4 Resulting in: 0.04, 0.04, 0.04, 0.04. The null hypothesis rejected if they are lower than a chosen false discovery rate. We used a false discovery rate of 0.05 to be consistent with the α value of 0.05 and the examples in Benjamini and Hochberg (1995). We made the used false discovery rate clearer in the Statistical Analysis subsection. None of these analyses modify the results and conclusions of our data.

Ref: https://www.jstor.org/stable/2346101

Controlling the False Discovery Rate: A Practical and Powerful Approach to Multiple Testing Yoav Benjamini and Yosef Hochberg Journal of the Royal Statistical Society. Series B (Methodological) Vol. 57, No. 1 (1995), pp. 289-300.

As requested, we have modified Figure 1 by adding the figure of the metabolic pathways (previously supplementary) in panel a.